# Minimum-Risk Recalibration of Classifiers

**Zeyu Sun**[*]
University of Michigan
zeyusun@umich.edu

**Dogyoon Song**[*]
University of Michigan
dogyoons@umich.edu

**Alfred Hero**
University of Michigan
hero@eecs.umich.edu

## Abstract

Recalibrating probabilistic classifiers is vital for enhancing the reliability and accuracy of predictive models. Despite the development of numerous recalibration algorithms, there is still a lack of a comprehensive theory that integrates calibration and sharpness (which is essential for maintaining predictive power). In this paper, we introduce the concept of minimum-risk recalibration within the framework of mean-squared-error (MSE) decomposition, offering a principled approach for evaluating and recalibrating probabilistic classifiers. Using this framework, we analyze the uniform-mass binning (UMB) recalibration method and establish a finite-sample risk upper bound of order $\tilde{O}(B/n + 1/B^2)$ where $B$ is the number of bins and $n$ is the sample size. By balancing calibration and sharpness, we further determine that the optimal number of bins for UMB scales with $n^{1/3}$, resulting in a risk bound of approximately $O(n^{-2/3})$. Additionally, we tackle the challenge of label shift by proposing a two-stage approach that adjusts the recalibration function using limited labeled data from the target domain. Our results show that transferring a calibrated classifier requires significantly fewer target samples compared to recalibrating from scratch. We validate our theoretical findings through numerical simulations, which confirm the tightness of the proposed bounds, the optimal number of bins, and the effectiveness of label shift adaptation.

## 1 Introduction

Generating reliable probability estimates alongside accurate class labels is crucial in classification tasks. A probabilistic classifier is considered "well calibrated" when its predicted probabilities closely align with the empirical frequencies of the corresponding labels [9]. Calibration is highly desirable, particularly in high-stakes applications such as meteorological forecasting [32, 34, 10, 15], econometrics [16], personalized medicine [25, 24], and natural language processing [37, 7, 11, 53]. Unfortunately, many machine learning algorithms lack inherent calibration [18].

To tackle this challenge, various methods have been proposed for designing post hoc recalibration functions. These functions are used to assess calibration error [52, 42, 17, 4], detect miscalibration [30], and provide post-hoc recalibration [50, 51, 18, 48, 29, 21]. Despite the rapid development of recalibration algorithms, there is still a lack of a comprehensive theory that encompasses both calibration and sharpness (retaining predictive power) from a principled standpoint. Furthermore, existing methods often rely on diverse calibration metrics [17, 4], and the selection of hyperparameters is often based on heuristic approaches [44, 20] without rigorous justifications. This highlights the

---

[*]Equal contribution.

37th Conference on Neural Information Processing Systems (NeurIPS 2023).

need for identifying an optimal metric to evaluate calibration, which can facilitate the development of a unified theory and design principles for recalibration functions.

In addition, the deployment of machine learning models to data distributions that differ from the training phase is increasingly common. These distribution shifts can occur naturally due to factors such as seasonality or other variations, or they can be induced artificially through data manipulation methods such as subsampling or data augmentation. Distribution shifts pose challenges to the generalization of machine learning models. Therefore, it becomes necessary to adapt the trained model to these new settings. One significant category of distribution shifts is label shift, where the marginal probabilities of the classes differ between the training and test sets while the class conditional feature distributions remain the same. With calibrated probabilistic predictions, label shift can be adjusted assuming access to class marginal probabilities [12]. However, miscalibration combined with label shift is common and remains a challenging problem [2, 14, 47, 41].

In this paper, we aim to address these issues in a twofold manner. Firstly, we develop a unified framework for recalibration that incorporates both calibration and sharpness in a principled manner. Secondly, we propose a composite estimator for recalibration in the presence of label shift that converges to the optimal recalibration. Our framework enables the adaptation of a classifier to the label-shifted domain in a sample-efficient manner.

## 1.1 Related work

**Recalibration algorithms.** Recalibration methods can be broadly categorized into parametric and nonparametric approaches. Parametric methods model the recalibration function in a parametric form and estimate the parameters using calibration data. Examples of parametric methods include Platt scaling [40], temperature scaling [18], Beta calibration [28], and Dirichlet calibration [27]. However, it has been reported that scaling methods are often less calibrated than supposed, and quantifying the degree of miscalibration can be challenging [29]. In contrast, nonparametric recalibration methods do not assume a specific parametric form for the recalibration function. These methods include histogram binning [50], isotonic regression [51], kernel density estimation [52, 42], splines [23], Gaussian processes [49], among others. Hybrid approaches, integrating both parametric and nonparametric techniques, have also been proposed. For instance, Kumar et al. [29] combine nonparametric histogram binning with parametric scaling to reduce variance and improve recalibration performance. Nevertheless, this hybrid approach is biased when its parametric assumptions fail. In this work, we consider a nonparametric histogram binning method called uniform-mass binning (UMB), which is asymptotically unbiased.

**Histogram binning method.** Histogram binning methods are widely used for recalibration due to their simplicity and adaptability. The binning schemes can be pre-specified (e.g., uniform-width binning [18]), data-dependent (e.g., uniform-mass binning [50]), or algorithm-induced [51]. When selecting a binning scheme, it is crucial to consider the trade-off between approximation and estimation. Coarser binning reduces estimation error (variance), leading to improved calibration, but at the expense of increased approximation error (bias), which diminishes sharpness. Thus, determining the optimal binning scheme and hyperparameters, such as the number of bins ($B$), remains an active area of research. [36] proposes a Bayesian binning method, but verifying the priors is often challenging. [44] suggests choosing the largest $B$ that preserves monotonicity, which is heuristic and computationally inefficient. [20] offers a heuristic for choosing the largest $B$ subject to a calibration constraint, lacking a quantitative characterization of sharpness. Our work builds upon the existing upper bounds for calibration risks of binning methods [29, 20] and derived upper bounds for a complementary risk component known as the sharpness risk. We quantitatively characterize the calibration-sharpness tradeoff, which yields an optimal choice for the number of bins that achieves the minimum risk.

**Adaptation to label shift.** Label shift presents a challenge in generalizing models trained on one distribution (source) to a different distribution (target). As such, adapting to label shift has received considerable attention in the literature [12, 45, 31, 3, 2, 14]. In practical scenarios, it is common to encounter model miscalibration and label shift simultaneously [47, 41]. Empirical observations have highlighted the crucial role of probability calibration in label shift adaptation [2, 13], which is justified by subsequent theories [14]. However, to the best of our knowledge, there has been no prior work that specifically addresses the recalibration with a limited amount of labeled data from the target distribution. Our theoretical analysis points out that using only target labels achieves risk bounds of the same order as the methods using only target features [31, 3, 14].

## 1.2 Contributions

This paper contributes to the theory of recalibration across three key dimensions.

Firstly, we develop a comprehensive theory for recalibration in binary classification by adopting the mean-squared-error (MSE) decomposition framework commonly used in meteorology and space weather forecasting [5, 33, 8, 46]. Our approach formulates the probability recalibration problem as the minimization of a specific risk function, which can be orthogonally decomposed into calibration and sharpness components.

Secondly, utilizing the aforementioned framework, we derive a rigorous upper bound on the finite-sample risk for uniform-mass binning (UMB) (Theorem 1). Furthermore, we minimize this risk bound and demonstrate that the optimal number of bins for UMB, balancing calibration and sharpness, scales on the order of $n^{1/3}$, yielding the risk bound of order $n^{-2/3}$, where $n$ denotes the sample size.

Lastly, we address the challenge of recalibrating classifiers for label shift when only a limited labeled sample from the target distribution is available, a challenging situation for a direct recalibration approach. We propose a two-stage approach: first recalibrating the classfier on the abundant source-domain data, and then transfering it to the label-shifted target domain. We provide a finite-sample guarantee for the risk of this composite procedure (Theorem 2). Notably, to control the risk under $\varepsilon$, our approach requires a much smaller sample size from the target distribution than a direct recalibration on the target sample ($\Omega(\varepsilon^{-1})$ vs. $\Omega(\varepsilon^{-3/2})$, cf. Remark 4).

## 1.3 Organization

This paper is organized as follows. In Section 2, we introduce notation and provide an overview of calibration and sharpness. Section 3 introduces the notion of minimum-risk recalibration by defining the recalibration risk that takes into account both calibration and sharpness. In Section 4, we describe the uniform-mass histogram binning method for recalibration and provide a risk upper bound with rate analysis. We extend our approach to handle label shift in Section 5. To validate our theory and framework, we present numerical experiments in Section 6. Finally, in Section 7, we conclude the paper with a discussion and propose future research directions.

## 2 Preliminaries

### 2.1 Notation

Let $\mathbb{N}$ and $\mathbb{R}$ denote the set of positive integers and the set of real numbers, respectively. For $n \in \mathbb{N}$, let $[n] := \{1, \ldots, n\}$. For $x \in \mathbb{R}$, let $\lfloor x \rfloor = \max\{m \in \mathbb{Z} : m \leq x\}$. For any finite set $\mathcal{S} = \{s_i : i \in [n]\} \subset \mathbb{R}$ and any $k \in [n]$, we let $s_{(k)}$ denote the $k$-th order statistic, which is the $k$-th smallest element in $\mathcal{S}$.

Let $(\Omega, \mathcal{E}, P)$ denote a generic probability space. For an event $A \in \mathcal{E}$, the indicator function $\mathbb{1}_A : \Omega \to \{0, 1\}$ is defined such that $\mathbb{1}_A(x) = 1$ if and only if $x \in A$. We write $A$ happens $P$-almost surely if $P(A) = 1$. For a probability measure $P$, define $\mathbb{E}_P$ as the expectation, with subscript $P$ omitted when the underlying probability measure is clear. For a probability measure $P$ and a random variable $X : \Omega \to \mathbb{R}^d$, let $P_X := P \circ X^{-1}$ be the probability measure induced by $X$.

Letting $f, g : \mathbb{R} \to \mathbb{R}$, we write $f(x) = O(g(x))$ as $x \to \infty$ if there exist $M > 0$ and $x_0 > 0$ such that $|f(x)| \leq M g(x)$ for all $x \geq x_0$. Likewise, we write $f(x) = \Omega(g(x))$ if $g(x) = O(f(x))$. We write $f(x) \asymp g(x)$ if $f(x) = O(g(x))$ and $g(x) = O(f(x))$. We write $f(x) = \tilde{O}(g(x))$ if there is $k \geq 1$ such that $f(x) = O(g(x) \log^k(g(x)))$.

### 2.2 Calibration and sharpness

Consider the binary classification problem; let $X \in \mathcal{X}$ and $Y \in \mathcal{Y} := \{0, 1\}$ denote the feature and label random variables. Letting $P$ denote a probability measure, we want to construct a function $f : \mathcal{X} \to \mathcal{Z} = [0, 1]$ that estimates the conditional probability, *i.e.*,

$$f(X) \approx P[Y = 1 \mid X]. \tag{1}$$

Since estimating the probability conditioned on the high dimensional $X$ is difficult, the notion of calibration captures the intuition of (1) in a weaker sense [33, 9, 19].

**Definition 1.** *A function $f : \mathcal{X} \to \mathcal{Z}$ is* (perfectly) calibrated *with respect to probability measure $P$, if*

$$f(X) = P\left[Y = 1 \mid f(X)\right] \qquad \text{P-almost surely.}$$

Calibration itself does not guarantee a useful predictor. For instance, a constant predictor $f(X) = \mathbb{E}Y$ is perfectly calibrated, but it does not change with the features. Such a predictor lacks sharpness [26], also known as resolution [35], another desired property which measures the variance in the target $Y$ explained by the probabilistic prediction $f(X)$.

**Definition 2.** *The* sharpness *of a function $f : \mathcal{X} \to \mathcal{Z}$ with respect to probability measure $P$ refers to the quantity*

$$\text{Var}\big(\mathbb{E}[Y \mid f(X)]\big) = \mathbb{E}\Big[\big(\mathbb{E}[Y \mid f(X)] - \mathbb{E}[Y]\big)^2\Big].$$

The following decomposition of the mean squared error (MSE) suggests why it is desirable for a classifier $f$ to be calibrated and have high sharpness; note that $\text{Var}[Y]$ is a quantity intrinsic to the problem, unrelated to $f$.

$$\text{MSE}(f) := \underbrace{\mathbb{E}\big[\big(Y - f(X)\big)^2\big]}_{\text{mean-squared error}} = \text{Var}[Y] - \underbrace{\text{Var}\big(\mathbb{E}\big[Y \mid f(X)\big]\big)}_{\text{sharpness}} + \underbrace{\mathbb{E}\big[\big(f(X) - \mathbb{E}[Y \mid f(X)]\big)^2\big]}_{\text{lack of calibration}}$$

$$(2)$$

## 3  Optimal recalibration

For an arbitrary predictor $f : \mathcal{X} \to \mathcal{Z}$, the aim of recalibration is to identify a post-processing function $h : \mathcal{Z} \to \mathcal{Z}$ such that $h \circ f$ is perfectly calibrated while maintaining the sharpness of $f$ as much as possible. The calibration and sharpness can be evaluated using the following two notions of risks. We suppress the dependency of risks on $f$ and $P$ when it leads to no confusion.

**Definition 3.** *Let $f : \mathcal{X} \to \mathcal{Z}$ and $h : \mathcal{Z} \to \mathcal{Z}$. The* calibration risk *of $h$ over $f$ under $P$ is defined as*

$$R^{\text{cal}}(h) = R_P^{\text{cal}}(h; f) := \mathbb{E}_P\left[\big(h \circ f(X) - \mathbb{E}_P\left[Y \mid h \circ f(X)\right]\big)^2\right]. \tag{3}$$

**Definition 4.** *Let $f : \mathcal{X} \to \mathcal{Z}$ and $h : \mathcal{Z} \to \mathcal{Z}$. The* sharpness risk *of $h$ over $f$ under $P$ is defined as*

$$R^{\text{sha}}(h) = R_P^{\text{sha}}(h; f) := \mathbb{E}_P\left[\big(\mathbb{E}_P\left[Y \mid h \circ f(X)\right] - \mathbb{E}_P\left[Y \mid f(X)\right]\big)^2\right]. \tag{4}$$

Note that the calibration risk $R^{\text{cal}}(h; f) = 0$ if and only if $h \circ f$ is perfectly calibrated, cf. Definition 1. The sharpness risk $R^{\text{sha}}(h; f)$ quantifies the decrement in sharpness of $f$ incurred by applying the recalibration map $h$, and $R^{\text{sha}}(h; f) = 0$ when $h$ is injective [29].

Next we define a comprehensive notion of risk that we will use to evaluate recalibration functions.

**Definition 5.** *Let $f : \mathcal{X} \to \mathcal{Z}$ and $h : \mathcal{Z} \to \mathcal{Z}$. The* recalibration risk *of $h$ over $f$ under $P$ is defined as*

$$R(h) = R_P(h; f) := \mathbb{E}_P\left[(h \circ f(X) - \mathbb{E}_P[Y \mid f(X)])^2\right]. \tag{5}$$

The following proposition shows that the recalibration risk can be decomposed into calibration risk and sharpness risk. The proof is deferred to Appendix A.

**Proposition 1** (Decomposition of recalibration risk)**.** *For any $f : \mathcal{X} \to \mathcal{Z}$ and any $h : \mathcal{Z} \to \mathcal{Z}$,*

$$R_P(h; f) = R_P^{\text{cal}}(h; f) + R_P^{\text{sha}}(h; f). \tag{6}$$

Note that $R_P(h) = 0$ if and only if $R_P^{\text{cal}}(h) = 0$ and $R_P^{\text{sha}}(h) = 0$. This happens if and only if $h \circ f$ is calibrated, and the recalibration $h$ preserves the sharpness of $f$ in predicting $Y$.

If $R(h) = 0$, then we call $h$ an *optimal recalibration function* (or *minimum-risk recalibration function*) of $f$ under $P$. Let $h_{f,P}^* : \mathcal{Z} \to \mathcal{Z}$ be the function

$$h_{f,P}^*(z) = \begin{cases} \mathbb{E}_P[Y \mid f(X) = z], & \text{if } z \in \text{supp } P_{f(X)}, \\ 0, & \text{if } z \notin \text{supp } P_{f(X)}. \end{cases} \tag{7}$$

Then $R(h^*_{f,P}) = 0$. Indeed, $h$ is a minimum-risk recalibration function of $f$ if and only if $h = h^*_{f,P}$ $P_Z$-almost surely.

**Problem 1** (Recalibration). *Suppose that we have a measurable function $f : \mathcal{X} \to \mathcal{Z}$ and a dataset $\mathcal{D} = \big\{(x_i, y_i) : i \in [n]\big\}$ that is an independent and identically distributed (IID) sample drawn from $P$. The goal of* recalibration *is to estimate a recalibration function $\hat{h} \approx h^*_{f,P}$ using $f$ and $\mathcal{D}$.*

Problem 1 can be viewed as a regression problem, where we estimate the function form of $\mathbb{E}[Y \mid Z]$ from data $\{(z_i, y_i) : i \in [n]\}$, where $z_i = f(x_i)$, $\forall i \in [n]$.

## 4 Recalibration via uniform-mass binning

### 4.1 Uniform-mass binning algorithm for recalibration

Given a dataset of prediction-label pairs $\{(z_i, y_i) \in \mathcal{Z} \times \mathcal{Y} : i \in [n]\}$, the histogram binning calibration method partitions $\mathcal{Z} = [0, 1]$ into a set of smaller bins, and estimate $\mathbb{E}[Y \mid Z]$ by taking the average in each bin. We consider the uniform mass binning, constructed using quantiles of predicted probabilities.

**Definition 6** (Uniform mass binning). *Let $S = \{z_i \in [0,1] : i \in [n]\}$. A binning scheme $\mathcal{B} = \{I_1, I_2, \ldots, I_B\}$ is the* uniform-mass binning *(UMB) scheme induced by $S$ if*

$$I_1 = [u_0, u_1], \qquad \text{and} \qquad I_b = (u_{b-1}, u_b] \;\; \forall b \in [B] \setminus \{1\}, \tag{8}$$

*where $u_0 = 0$, $u_B = 1$, and $u_b = z_{(\lfloor nb/B \rfloor)}$ for $b \in [B-1]$.*

For our subsequent discussions, we make the following assumptions on the distribution of $Y$ and $Z$:

(A1) The cumulative distribution function of $Z$, denoted by $F_Z$, is absolutely continuous.

(A2) $h^*_{f,P}$, defined in (7), is monotonically non-decreasing on $\operatorname{supp} P_Z$.

(A3) There exists $K > 0$ such that if $z_1 \leq z_2$, then $h^*_{f,P}(z_2) - h^*_{f,P}(z_1) \leq K \cdot \big(F_Z(z_2) - F_Z(z_1)\big)$.

Assumption (A1) is made for the sake of analytical convenience without loss of generality; see discussions in [20, Appendix C]. An important implication of Assumption (A1) is that all intervals in $\mathcal{B}$ are non-empty $P_Z$-almost surely if $S = \{Z_i : i \in [n]\}$ are IID under $P_Z$. Assumption (A2) makes sure $Z$ is informative to preserve the rankings of $P[Y \mid Z]$ [51]. Lastly, Assumption (A3) posits that $Z$ is sufficiently informative that $P[Y = 1 \mid Z = z]$ does not change too rapidly in any interval $I$ where $P_Z(I)$ is small. This assumption is mild but not trivial; see Appendix B.2.

Now we describe how to construct a recalibration function $\hat{h}$ using UMB.

**Algorithm.**

1. Given $\{(z_i, y_i) : i \in [n]\}$, and $B \in \mathbb{N}$, let $\mathcal{B} = \{I_1, I_2, \ldots, I_B\}$ be the UMB scheme of size $B$ induced by $\{z_i : i \in [n]\}$.

2. Let $\hat{h} = \hat{h}_{\mathcal{B}} : \mathcal{Z} \to \mathcal{Z}$ be a function such that

$$\hat{h}(z) = \sum_{I \in \mathcal{B}} \hat{\mu}_I \cdot \mathbb{1}_I(z) \quad \text{where} \quad \hat{\mu}_I := \frac{\sum_{i=1}^n y_i \cdot \mathbb{1}_I(z_i)}{\sum_{i=1}^n \mathbb{1}_I(z_i)}, \;\; \forall I \in \mathcal{B}. \tag{9}$$

### 4.2 Theoretical guarantees

Here we establish a high-probability upper bound on the recalibration risk $R(\hat{h})$ for the UMB estimator, $\hat{h}$ defined in (9), which converges to 0 as the sample size $n$ increases to $\infty$.

**Theorem 1.** *Let $P$ be a probability measure and $f : \mathcal{X} \to \mathcal{Z}$ be a measurable function. Suppose that (A1) & (A2) hold. Let $\mathcal{B}$ be the UMB scheme induced by an IID sample of $f(X)$, and let $\hat{h} = \hat{h}_{\mathcal{B}}$ be the recalibration function based on $\mathcal{B}$, cf. (9). Then there exists a universal constant $c > 0$ such that for any $\delta \in (0, 1)$, if $n \geq c \cdot |\mathcal{B}| \log\big(\frac{2|\mathcal{B}|}{\delta}\big)$, then with probability at least $1 - \delta$,*

$$R^{\mathrm{cal}}(\hat{h}) \leq \left( \sqrt{\frac{\log(4|\mathcal{B}|/\delta)}{2(\lfloor n/|\mathcal{B}| \rfloor - 1)}} + \frac{1}{\lfloor n/|\mathcal{B}| \rfloor} \right)^2, \quad \text{and} \quad R^{\mathrm{sha}}(\hat{h}) \leq \begin{cases} \frac{2}{|\mathcal{B}|}; \\ \frac{8K^2}{|\mathcal{B}|^2} & \text{if (A3) holds.} \end{cases}$$

**Remark 1.** *We note that the upper bound on $R^{\text{cal}}(\hat{h})$ in Theorem 1 coincides with the result presented in [20] up to a constant factor in the failure probability. However, Theorem 1 provides an additional upper bound on $R^{\text{sha}}(\hat{h})$, thereby effectively managing the overall recalibration risk $R(\hat{h})$.*

**Optimal choice of the number of bins $|\mathcal{B}|$.** Based on Theorem 1, for any fixed sample size $n$, as the number of bins $B = |\mathcal{B}|$ increases, the calibration risk bound increases, while the sharpness risk bound decreases. This trade-off suggests we may get an optimal number of bins $B$ by minimizing the upper bound for the overall risk, $R(\hat{h}) = R^{\text{cal}}(\hat{h}) + R^{\text{sha}}(\hat{h})$.

For the simplicity of our analysis, we assume $n$ is divisible by $B$, and Assumption (A3) holds. Firstly, observe that $n/B \geq 2$, and thus, $n/B - 1 \geq n/(2B)$. Also, since $B \geq 1$ and $\delta < 1$, $\log(\frac{4B}{\delta}) \geq 1$, it follows that $\frac{1}{n/B} \leq \sqrt{\frac{1}{2(n/B-1)} \log\left(\frac{4B}{\delta}\right)} \leq \sqrt{\frac{1}{n/B} \log\left(\frac{4B}{\delta}\right)}$. Thus, we obtain a simplified risk bound $R(\hat{h}) \leq \zeta(B; n, \delta)$ where $\zeta(B; n, \delta) := \frac{4B}{n} \log\left(\frac{4B}{\delta}\right) + \frac{8K^2}{B^2}$. With $\log B$ terms ignored, minimizing $\zeta(B; n, \delta)$ over $B$ yields optimal $B^*$ and resulting risk bounds, respectively:

$$B^* \asymp n^{1/3} \cdot \log^{-1/3}(1/\delta), \tag{10}$$

$$R(\hat{h}) \asymp R^{\text{cal}}(\hat{h}) \asymp R^{\text{sha}}(\hat{h}) = O\left(B^{*-2}\right) = O\left(n^{-2/3} \cdot \log^{2/3}(1/\delta)\right). \tag{11}$$

Furthermore, the asymptotic risk bound in (11) implies that $R(\hat{h})$, $R^{\text{cal}}(\hat{h})$, and $R^{\text{sha}}(\hat{h})$ are all bounded by $\varepsilon$ with high probability if the sample size

$$n = \Omega(\varepsilon^{-3/2}). \tag{12}$$

**Comparison with the hybrid method [29].** Kumar et al. [29] proposed a hybrid recalibration approach that involves fitting a recalibration function in a parametric family $\mathcal{H}$, which is then discretized using UMB. Note that the hypothesis class $\mathcal{H}$ may or may not include the optimal recalibration function $h_{f,P}^*$. If $h_{f,P}^* \in \mathcal{H}$, using a similar analysis as above, we derive their high probability risk bound as $O\left(\frac{1}{n} \log \frac{B}{\delta} + \frac{1}{B^2}\right)$ under Assumption (A3), which achieves $\tilde{O}(n^{-1})$ when $B^* \asymp \sqrt{n}$, exhibiting a faster decay than our $R(\hat{h}) = O(n^{-2/3})$. While the faster rate is anticipated from employing parametric methods, we note that when $h_{f,P}^* \notin \mathcal{H}$, their method exhibits inherent bias (approximation error) induced by parametric function fitting, whereas our method is asymptotically unbiased. This distinction is corroborated by numerical simulations in Appendix E.2.

**Proof sketch of Theorem 1.** First, we demonstrate that the uniform-mass binning scheme $\mathcal{B} = \{I_b\}_{b=1}^B$ satisfies two regularity conditions with high probability, when the sample size $n$ is not too small. Specifically, we show that (i) $\mathcal{B}$ is 2-well-balanced [29] with respect to $f(X)$, resulting in $B$ bins having comparable probabilities (Lemma 3); and that (ii) the empirical mean in each bin of $\mathcal{B}$ uniformly concentrates to the population conditional mean of $Y$ conditioned on $f(X)$ being contained within the bin (Lemma 4). Thereafter, we prove that if $\mathcal{B}$ satisfies these two properties, then the calibration risk $R^{\text{cal}}$ and the sharpness risk $R^{\text{sha}}$ can be upper bounded as stated in Theorem 1; see Lemmas 5 and 6 (or Lemma 7 when (A3) holds). The detailed proof is in Appendix B.

## 5 Recalibration under label shift

This section extends the results from Section 4 to address label shift. In Section 5.1, we introduce the label shift assumption (Definition 7) and reframe the recalibration problem accordingly. We show that the optimal recalibration function in this context can be expressed as a composition of the optimal recalibration function (cf. Section 3) and a shift correction function. Building on this observation, we propose a two-stage estimator in Section 5.2, where each stage estimates one of the component functions. The composite estimator's overall performance is supported by theoretical guarantees.

### 5.1 Revisiting the problem formulation

Let $P$ and $Q$ denote the probability measures of the source and the target domains, respectively. We assume $P$ and $Q$ satisfy the label shift assumption defined below.

**Definition 7** (Label shift). *Probability measures $P$ and $Q$ are said to satisfy the* label shift *assumption if the following two conditions are satisfied:*

(B1) $P[X \in B \mid Y = k] = Q[X \in B \mid Y = k]$ *for all* $k \in \{0, 1\}$ *and all* $B \in \mathcal{B}(\mathcal{X})$.

(B2) $P[Y = 1] \in (0, 1)$ *and* $Q[Y = 1] \in (0, 1)$.

According to Condition (B1), the class conditional distributions remain the same, while the marginal distribution of the classes may change. Condition (B2) requires all classes to be present in the source population, which is a standard regularity assumption in the discussion of label shift [31, 14]; it also posits the presence of every class in the target population.

**Optimal recalibration under label shift.** Under the label shift assumption between $P$ and $Q$, we define the label shift correction function $g^* : \mathcal{Z} \to \mathcal{Z}$ such that

$$g^*(z) = \frac{w_1^* z}{w_1^* z + w_0^*(1 - z)} \qquad \text{where} \qquad w_k^* = \frac{Q[Y = k]}{P[Y = k]}, \quad \forall k \in \{0, 1\}. \tag{13}$$

The conditional probabilities under $P$ and $Q$ can be related [45] as follows:

$$Q[Y = 1 \mid X \in B] = g^*\Big(P[Y = 1 \mid X \in B]\Big), \qquad \forall B \in \mathcal{B}(\mathcal{X}). \tag{14}$$

Recall that the optimal recalibration function for a predictor $f : \mathcal{X} \to \mathcal{Z}$ under probability measure $P$ is defined as $h_{f,P}^*(z) = P[Y = 1 \mid f(X) = z]$; see (7). In the presence of a label shift between $P$ and $Q$, we may write the optimal recalibration function for $f$ under $Q$ as

$$h_{f,Q}^* = g^* \circ h_{f,P}^* \tag{15}$$

because $h_{f,Q}^*(z) \overset{(a)}{=} Q[Y = 1 \mid f(X) = z] \overset{(b)}{=} g^*(P[Y = 1 \mid f(X) = z]) \overset{(c)}{=} \left(g^* \circ h_{f,P}^*\right)(z)$, where (a) and (c) follows from the definition of $h_{f,P}^*$ and (b) is due to (14).

Recalling the definition of the risk $R_P(h; f)$ from (5), we observe that $R_Q(h_{f,Q}^*; f) = 0$, which is consistent with the risk characterization of the optimal recalibration. Our goal is to estimate the optimal recalibration function $h_{f,Q}^* = g^* \circ h_{f,P}^*$ from data.

**Problem 2** (Recalibration under label shift). *Suppose that we have a measurable function $f : \mathcal{X} \to \mathcal{Z}$ and two IID datasets $\mathcal{D}_P = (x_i, y_i)_{i=1}^{n_P} \sim P$ and $\mathcal{D}_Q = (x_i', y_i')_{i=1}^{n_Q} \sim Q$. The goal of* recalibration under label shift *is to estimate $\hat{h} \approx h_{f,Q}^*$ using $f$, $\mathcal{D}_P$ and $\mathcal{D}_Q$.*

**Remark 2.** *The source (training) dataset $\mathcal{D}_P$ may not be accessible due to privacy protections, proprietary data, or practical constraints, as is often the case when recalibrating a pre-trained black box classifier to new data. In these cases, it suffices to have estimates of the recalibration function $h_{f,P}^*$ and the marginal probabilities $P[Y = k]$, $k \in \{0, 1\}$ under $P$, for our method and analysis.*

## 5.2 Two-stage recalibration under label shift

**Method.** We propose a composite estimator of $h_{f,Q}^* = g^* \circ h_{f,P}^*$, which comprises two estimators $\hat{g} \approx g^*$ and $\hat{h}_P \approx h_{f,P}^*$. Here we describe a procedure to produce this composite estimator.

1. Use $\mathcal{D}_P$ to construct $\hat{h}_P : \mathcal{Z} \to \mathcal{Z}$, the estimated recalibration function (9) (for $f$ under $P$).
2. Use $\mathcal{D}_P$ and $\mathcal{D}_Q$ to construct $\hat{g} : \mathcal{Z} \to \mathcal{Z}$ such that

$$\hat{g}(z) = \frac{\hat{w}_1 z}{\hat{w}_1 z + \hat{w}_0(1 - z)} \qquad \text{where} \qquad \hat{w}_k = \frac{\hat{Q}[Y = k]}{\hat{P}[Y = k]}, \quad \forall k \in \{0, 1\}, \tag{16}$$

   where $\hat{P}[Y = k] := \frac{1}{|\mathcal{D}_P|} \sum_{i=1}^{|\mathcal{D}_P|} \mathbb{1}[y_i = k]$ and $\hat{Q}[Y = k] := \frac{1}{|\mathcal{D}_Q|} \sum_{i=1}^{|\mathcal{D}_Q|} \mathbb{1}[y_i' = k]$ are the empirical estimates of the class marginal probabilities.
3. Let

$$\hat{h}_Q = \hat{g} \circ \hat{h}_P. \tag{17}$$

Note that the recalibration estimator $\hat{h}_P$ (Step 1) remains the same with that in Section 4.1. Furthermore, the shift correction estimator $\hat{g}$ (Step 2) is a plug-in estimator of the label shift correction function $g^*$ in (14) based on the estimated weights, $\hat{w}_1$ and $\hat{w}_0$.

**Theory.** We present a recalibration risk upper bound for the proposed two-stage estimator. We let $p_k := P[Y = k]$, $q_k := Q[Y = k]$, and $w_k^* = \frac{q_k}{p_k}$ for $k \in \{0, 1\}$. Moreover, we let $p_{\min} := \min_k p_k$, $q_{\min} := \min_k p_k$, $w_{\min}^* := \min_k w_k^*$ and $w_{\max}^* := \max_k w_k^*$.

**Theorem 2** (Convergence of $\hat{h}_Q$)**.** *Let $P, Q$ be probability measures and let $f : \mathcal{X} \to \mathcal{Z}$ be a measurable function. Let $\mathcal{D}_P \sim P$ be an IID sample of size $n_P$ and $\mathcal{D}_Q \sim Q$ be an IID sample of size $n_Q$. Suppose that Assumptions (B1) & (B2) hold. Let $\mathcal{B}$ be the UMB scheme induced by $\mathcal{D}_P$. Let $\hat{h}_P = \hat{h}_{P,\mathcal{B}}$ be the recalibration function* (9) *based on $\mathcal{B}$, and let $\hat{g}$ denote the shift correction function as defined in* (16). *Then*

$$R_Q(\hat{g} \circ \hat{h}_P) \leq 2 \left\{ \left( \frac{\rho_0 - \rho_1}{\rho_0 + \rho_1} \right)^2 + \frac{{w_{\max}^*}^3}{{w_{\min}^*}^2} \cdot R_P(\hat{h}_P; f) \right\} \qquad where \qquad \rho_k := \frac{\hat{w}_k}{w_k^*}, \quad k \in \{0, 1\}. \tag{18}$$

*Furthermore, suppose that (A1), (A2) & (A3) hold. Then there exists a universal constant $c > 0$ such that for any $\delta \in (0, 1)$, if*

$$n_P \geq \max \left\{ c, \frac{27}{p_{\min}} \right\} \cdot |\mathcal{B}| \log \left( \frac{4|\mathcal{B}|}{\delta} \right) \qquad and \qquad n_Q \geq \frac{27}{q_{\min}} \log \left( \frac{16}{\delta} \right),$$

*then with probability at least $1 - \delta$,*

$$R_Q(\hat{g} \circ \hat{h}_P) \leq 2 \frac{{w_{\max}^*}^3}{{w_{\min}^*}^2} \cdot \left\{ \left( \sqrt{\frac{1}{2(\lfloor n_P/|\mathcal{B}| \rfloor - 1)} \log \left( \frac{8|\mathcal{B}|}{\delta} \right)} + \frac{1}{\lfloor n_P/|\mathcal{B}| \rfloor} \right)^2 + \frac{8K^2}{|\mathcal{B}|^2} \right\}$$
$$+ 54 \max \left\{ \frac{1}{p_{\min} \cdot n_P}, \frac{1}{q_{\min} \cdot n_Q} \right\} \cdot \log \left( \frac{16}{\delta} \right). \tag{19}$$

**Remark 3.** *Note that $\rho_k \to 1$ as $n_P, n_Q \to \infty$, and thus, the upper bound* (18) *reduces to $2 \frac{{w_{\max}^*}^3}{{w_{\min}^*}^2} \cdot R_P(\hat{h}_P; f)$. Moreover, when $P = Q$, we have $w_{\min}^* = w_{\max}^* = 1$, and this further simplifies to the recalibration risk without label shift, up to multiplicative constant $2$.*

**Remark 4** (Target sample complexity)**.** *Assume that $n_P \geq n_Q$ and the number of bins satisfies $|\mathcal{B}| \asymp n_P^{1/3}$. Then* (19) *implies $R_Q(\hat{g} \circ \hat{h}_P; f) = O(n_P^{-2/3} + n_Q^{-1})$. This result indicates that the proposed recalibration method using* (17) *requires a significantly smaller target sample size $n_Q = \Omega(\varepsilon^{-1})$ to control the risk, as compared to $n_Q = \Omega(\varepsilon^{-3/2})$ in* (12).

**Remark 5** (Comparison with label shift using unlabeled target data)**.** *When the source sample size $n_P$ is sufficiently large, we achieve a risk of $R_Q(\hat{g} \circ \hat{h}_P) = O(n_Q^{-1})$ with high probability. It is important to note that in this scenario, we only utilize the labels from the target sample to address label shift. Remarkably, the same rate applies when employing the algorithms proposed in [31, 3, 14], which solely rely on features from the target sample. For a proof sketch, please refer to Appendix D.*

# 6 Numerical experiments

In this section, we present the results of our numerical simulations conducted to validate and reinforce the theoretical findings discussed earlier. The simulations are based on a family of recalibration functions called beta calibration [28]: $\mathcal{H}_{\text{beta}} = \{h_{\text{beta}}(z; a, b, c) : a \geq 0, b \geq 0, c \in \mathbb{R}\}$, where $h_{\text{beta}}(\cdot; a, b, c) : [0, 1] \to [0, 1]$ is defined as

$$h_{\text{beta}}(z; a, b, c) = \frac{1}{1 + 1/\left( e^c \frac{z^a}{(1-z)^b} \right)}. \tag{20}$$

In addition, consider the joint distributions $\mathcal{D}(\pi)$ of $X$ and $Y$, where $Y \sim \text{Bernoulli}(\pi)$, $X \mid Y = 0 \sim N(-2, 1)$, and $X \mid Y = 1 \sim N(2, 1)$, and a pre-trained probabilistic classifier $f(x) = \sigma(x) := 1/(1 + e^{-x})$. To accommodate the limitations of space, we summarize the results in Figure 1, 2, 3, and Table 1, 2, providing a concise overview. Detailed information about the simulation settings, implementation details, and further experimental results and discussions can be found in Appendix E. Our simulation code is available at `https://github.com/ZeyuSun/calibration_label_shift`.

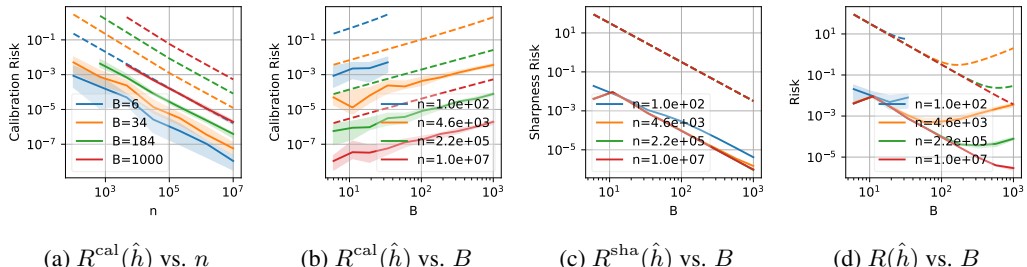

| (a) $R^{\text{cal}}(\hat{h})$ vs. $n$ | (b) $R^{\text{cal}}(\hat{h})$ vs. $B$ | (c) $R^{\text{sha}}(\hat{h})$ vs. $B$ | (d) $R(\hat{h})$ vs. $B$ |
|---|---|---|---|

Figure 1: Medians (solid lines) and 10-90 percentile ranges (shaded areas) of quadrature estimates of population risks over 10 realizations and theoretical risk upper bounds ($\delta = 0.1$) (dashed lines) for various $n$ and $B$. **(a)-(c)** The empirical rates, $R^{\text{cal}} = O(n^{-0.99}B^{0.98})$ and $R^{\text{sha}} = O(B^{-1.83})$, align with theoretically predicted rates, $\tilde{O}(B/n)$ and $O(B^{-2})$, in Thm. 1. **(d)** The empirically observed $R$ and our upper bound exhibit similar trends as a function of $B$.

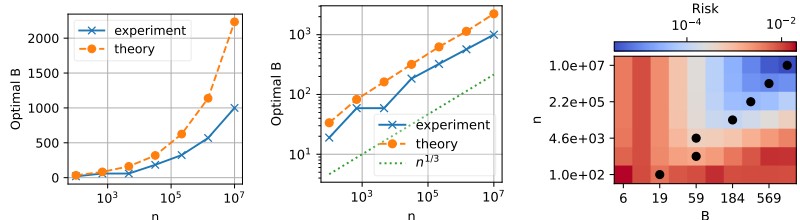

Figure 2: The optimal number of bins $B$ for different sample sizes $n$ plotted in linear scale (*left*) and log scale (*middle*), and the population risk for various combinations of $n$ and $B$, with the optimal $B$ marked by black dots (*right*). Note that the risk surface is relatively smooth around its minimum, suggesting the robustness of the optimal $B$.

Table 1: 90%-quantiles of the risks of Platt scaling [40], the hybrid method [29], uniform-width binning (UWB) [18], and uniform-mass binning (UMB) over 100 random calibration datasets drawn from $Z \sim \text{Uniform}[0, 1]$, and **(a)** $Y \mid Z \sim \text{Bernoulli}(h_{\text{beta}}(z; 4, 4, 0))$, or **(b)** $Y \mid Z \sim \text{Bernoulli}(h_{\text{beta}}(z; 0.1, 4, 0))$. While Platt scaling and the hybrid method achieve lower $R$ under the correct parametric assumption, UWB and UMB may outperform when the parametric assumption fails.

(a) Correct parametric assumption

| Metric ($\times 10^{-3}$) | $R^{\text{cal}}$ | $R^{\text{sha}}$ | $R$ | MSE |
|---|---|---|---|---|
| Platt | 0.122 | **0.000** | **0.122** | **10.338** |
| Hybrid | **0.119** | 0.212 | 0.315 | 10.532 |
| UWB | 0.661 | 0.194 | 0.855 | 11.071 |
| UMB | 0.647 | 0.212 | 0.839 | 11.055 |

(b) Misspecified parametric assumption

| Metric ($\times 10^{-3}$) | $R^{\text{cal}}$ | $R^{\text{sha}}$ | $R$ | MSE |
|---|---|---|---|---|
| Platt | 3.682 | **0.000** | 3.682 | 21.994 |
| Hybrid | 3.117 | 0.251 | 3.360 | 21.672 |
| UWB | 0.572 | 0.238 | 0.810 | 19.122 |
| UMB | **0.560** | 0.251 | **0.797** | **19.109** |

Table 2: Risks under label shift from $\mathcal{D}(0.5)$ to $\mathcal{D}(0.1)$, with $n_P = 10^3$ and $n_Q = 10^2$. Standard deviations are computed from 10 random realizations. LABEL-SHIFT, only applying an injective $\hat{g}$, achieves $R^{\text{sha}} = 0$ but incurs high $R^{\text{cal}}$. SOURCE, recalibrated with $B = n_P^{1/3}$ on $\mathcal{D}_P$, incurs high $R^{\text{cal}}$. TARGET, recalibrated with $B = n_Q^{1/3}$ on $\mathcal{D}_Q$, incurs low $R^{\text{cal}}$. Our proposed COMPOSITE achieves the lowest nonzero $R^{\text{sha}}$ and lowest $R^{\text{cal}}$.

| Method | Estimator | $R^{\text{cal}}$ | $R^{\text{sha}}$ | $R$ | MSE |
|---|---|---|---|---|---|
| SOURCE | $\hat{h}_P$ | 0.016±0.005 | 0.0032±0.0014 | 0.019±0.006 | 0.029±0.006 |
| TARGET | $\hat{h}_Q^{\text{target}}$ | 0.0020±0.0025 | 0.049±0.006 | 0.051±0.006 | 0.060±0.006 |
| LABEL-SHIFT | $\hat{g}$ | 0.026±0.006 | **0** | 0.026±0.006 | 0.035±0.006 |
| COMPOSITE | $\hat{h}_Q$ | **0.00019±0.00017** | 0.0032±0.0014 | **0.0034±0.0013** | **0.0127±0.0013** |

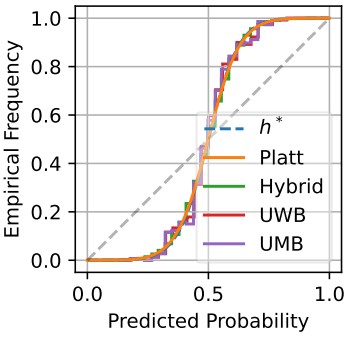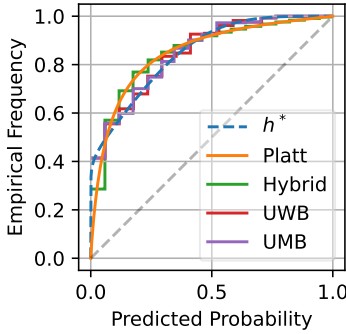

(a) Correct parametric assumption             (b) Misspecified parametric assumption

Figure 3: Optimal recalibration function $h^*$ and recalibration function estimates by Platt Scaling [40], the hybrid method [29], UWB, and UMB when the parametric assumption is **(a)** correct and **(b)** misspecified. UMB traces the $h^*$ in both cases, whereas the hybrid method traces Platt scaling, exhibiting an intrinsic bias from $h^*$ in **(b)**.

## 7  Discussion

This paper presents a comprehensive theory for recalibration, considering both calibration and sharpness within the mean-squared-error (MSE) decomposition framework. We use this framework to quantify the optimal calibration-sharpness balance and establish a rigorous upper bound on the finite-sample risk for uniform-mass binning (UMB). Additionally, we address the challenge of recalibration under label shift with limited access to labeled target data. Our proposed two-stage approach effectively estimates the recalibration function using ample data from the source domain and adjusts for the label shift using target domain data. Importantly, our findings suggest that transferring a calibrated classifier requires a significantly smaller target sample than recalibrating from scratch on the new domain. Numerical simulations confirm the tightness of the finite sample bounds, validate the optimal number of bins, and demonstrate the effectiveness of the label shift adaptation.

In concluding this paper, we identify several promising directions for future research.

**Relaxation of the assumptions**  It would be worthwhile to explore whether or not the assumptions made in our analysis could be relaxed. For instance, the widely adopted monotonicity assumption (A2) and its variants [51, 52, 43, 44] may not hold in real-world settings. Thus, exploring potential relaxations of this assumption is valuable. In addition, a mild but non-trivial smoothness assumption (A3) (c.f. Remark 7 and 8) is introduced to obtain a tight sharpness risk upper bound (c.f. Remark 9); investigating its practical implications and exploring potential relaxations could be interesting future work.

**Calibration-sharpness framework analysis**  Applying our framework to analyze recalibration methods beyond UMB, such as isotonic regression [51] and kernel density estimation [52], can offer further insights into their performance and properties, providing guidance to practitioners in selecting suitable algorithms based on specific conditions and requirements.

**Multiclass probability recalibration**  The concept of calibration considered in this work extends to multi-class classification settings, known as canonical calibration [48]. Weaker, but more tractable notions of multi-class calibration have also been explored in the literature [48, 21]. While partition-based methods, as multi-class extensions of binning methods, are known to have consistency [48] and vanishing calibration error [41], establishing upper bounds for their sharpness risk remains difficult. Additionally, designing a partition scheme in a multidimensional space is a challenging task [21]; the interplay between calibration and sharpness could potentially guide the development of partition strategies that balance both aspects in multi-class classification.

**Applications to real-world data**  Investigating the calibration-sharpness tradeoff in real-world applications, which often involve multiple classes, presents an interesting challenge. It is crucial to develop effective estimators for both calibration risk and sharpness risk in such scenarios. While estimators for calibration risk exist (e.g., binning-based and KDE-based) [6, 29, 52, 42] and a lower bound for sharpness risk has been established [39], a direct estimator of sharpness risk is still lacking.

## Acknowledgments and Disclosure of Funding

The research in this paper was partially supported by ARO grants W911NF-23-1-0343 and W911NF-19-1-0269 and by DOE grant DE-NA0003921.

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

# A  Proof of Proposition 1

*Proof of Proposition 1.* First of all, we recall the definition of the two risks from (3) and (4):

$$R^{\mathrm{cal}}(h; f) = \mathbb{E}\left[\left(f(X) - \mathbb{E}\left[Y \mid f(X)\right]\right)^2\right]$$

$$R^{\mathrm{sha}}(h; f) = \mathbb{E}\left[\left(\mathbb{E}\left[Y \mid h \circ f(X)\right] - \mathbb{E}\left[Y \mid f(X)\right]\right)^2\right].$$

To keep our notation concise, we use $Z = f(X)$ as a shorthand notation, and also let $Y_Z := \mathbb{E}[Y \mid Z]$ and $Y_{h(Z)} = \mathbb{E}[Y \mid h(Z)]$ throughout this proof. We can decompose the recalibration risk from Definition 5:

$$
\begin{aligned}
R(h) &= \mathbb{E}[(h(Z) - Y_Z)^2] \\
&= \mathbb{E}[(h(Z) - Y_{h(Z)} + Y_{h(Z)} - Y_Z)^2] \\
&= \mathbb{E}[(h(Z) - Y_{h(Z)})^2] + \mathbb{E}[(Y_{h(Z)} - Y_Z)^2] + 2\mathbb{E}[(h(Z) - Y_{h(Z)})(Y_{h(Z)} - Y_Z)] \\
&= \mathbb{E}[(h(Z) - Y_{h(Z)})^2] + \mathbb{E}[(Y_{h(Z)} - Y_Z)^2] + 2\mathbb{E}[\mathbb{E}[(h(Z) - Y_{h(Z)})(Y_{h(Z)} - Y_Z) \mid h(Z)]] \\
&= \mathbb{E}[(h(Z) - Y_{h(Z)})^2] + \mathbb{E}[(Y_{h(Z)} - Y_Z)^2] + 2\mathbb{E}[(h(Z) - Y_{h(Z)})(Y_{h(Z)} - \mathbb{E}[Y_Z \mid h(Z)])] \\
&= \mathbb{E}[(h(Z) - Y_{h(Z)})^2] + \mathbb{E}[(Y_{h(Z)} - Y_Z)^2] + 2\mathbb{E}[(h(Z) - Y_{h(Z)})(Y_{h(Z)} - Y_{h(Z)})] \\
&= \underbrace{\mathbb{E}[(h(Z) - Y_{h(Z)})^2]}_{R^{\mathrm{cal}}(h)} + \underbrace{\mathbb{E}[(Y_{h(Z)} - Y_Z)^2]}_{R^{\mathrm{sha}}(h)}.
\end{aligned}
$$

$\square$

# B  Proof of Theorem 1

In this section, we present a proof of Theorem 1. Let $(Y, Z) \in \mathcal{Y} \times \mathcal{Z}$ be random variables that admits a joint distribution $P_{Y,Z}$, which we assume to be fixed throughout this section. Let $S = \{(y_i, z_i) \in \mathcal{Y} \times \mathcal{Z} : i \in [n]\}$ and let $\mathcal{B} = \{I_1, I_2, \ldots, I_B\}$ be the uniform-mass binning scheme (cf. Definition 6) of size $B$ induced by ($z_i$'s in) $S$. Note that if $S$ is a random sample from $P_{Y,Z}$, then the binning scheme $\mathcal{B}$ induced by $S$ is also a random variable following a derived distribution. To facilitate our analysis, we introduce the notion of well-balanced binning.

**Definition 8** (Well-balanced binning; [29])**.** *Let $B \in \mathbb{N}$, let $Z$ be a random variable that takes value in $[0,1]$, and let $\alpha \in \mathbb{R}$ such that $\alpha \geq 1$. A binning scheme $\mathcal{B}$ of size $B$ is $\alpha$-well-balanced with respect to $Z$ if*

$$\frac{1}{\alpha B} \leq P[Z \in I_b] \leq \frac{\alpha}{B}, \qquad \forall b \in [B].$$

In addition, we define two (parameterized families of) Boolean-valued functions $\Phi_{\mathrm{balance}}$ and $\Phi_{\mathrm{approx}}$ as follows: for any binning scheme $\mathcal{B}$,

$$\forall \alpha \in \mathbb{R}, \quad \Phi_{\mathrm{balance}}(\mathcal{B}; \alpha) := \mathbb{1}\left\{\frac{1}{\alpha|\mathcal{B}|} \leq P\left[Z \in I\right] \leq \frac{\alpha}{|\mathcal{B}|}, \ \forall I \in \mathcal{B}\right\}, \tag{21}$$

$$\forall \varepsilon \in \mathbb{R}, \quad \Phi_{\mathrm{approx}}(\mathcal{B}; \varepsilon) := \mathbb{1}\left\{\max_{I \in \mathcal{B}} |\hat{\mu}_I - \mu_I| \leq \varepsilon\right\}, \tag{22}$$

where $\mathbb{1}(A) = 1$ if and only if the predicate $A$ is true, and for each interval $I \in \mathcal{B}$,

$$\hat{\mu}_I = \frac{\sum_{i=1}^n y_i \cdot \mathbb{1}_I(z_i)}{\sum_{i=1}^n \cdot \mathbb{1}_I(z_i)} \qquad \text{and} \qquad \mu_I = \mathbb{E}_{(Y,Z) \sim P_{Y,Z}}\left[Y \cdot \mathbb{1}_I(Z)\right]. \tag{23}$$

Note that if $\Phi_{\mathrm{balance}}(\mathcal{B}; \alpha) = 1$ for $\alpha \geq 1$, then $\mathcal{B}$ is $\alpha$-well-balanced with respect to $Z$ (cf. Definition 8). Also, if $\Phi_{\mathrm{approx}}(\mathcal{B}; \varepsilon) = 1$ for $\varepsilon \geq 0$, then the conditional empirical mean of $Y$ in each bin $I \in \mathcal{B}$ approximates the conditional expectation with error at most $\varepsilon$, uniformly for all bins.

The rest of this section is organized as follows. In Section B.1, we ensure that for an appropriate choice of $\alpha, \varepsilon \in \mathbb{R}$, it holds with high probability (with respect to the randomness in $\mathcal{B}$) that $\Phi_{\mathrm{balance}}(\mathcal{B}; \alpha) = \Phi_{\mathrm{approx}}(\mathcal{B}; \varepsilon) = 1$. In Section B.2, we establish upper bounds on the reliability risk $R^{\mathrm{cal}}$ and the sharpness risk $R^{\mathrm{sha}}$ under the premise that $\Phi_{\mathrm{balance}}(\mathcal{B}; \alpha) = \Phi_{\mathrm{approx}}(\mathcal{B}; \varepsilon) = 1$. Finally, in Section B.3, we conclude the proof of Theorem 1 by combining these results together.

## B.1 High-probability certification of the conditions

**Well-balanced binning scheme.** First of all, we observe that the uniform-bass binning scheme $\mathcal{B}$ induced by an IID random sample from $P_{Y,Z}$ is 2-well-balanced with high probability, if the sample size is sufficiently large. Here we paraphrase a result from [29] in our language.

**Lemma 3** ([29, Lemma 4.3]). *Let $S = \{Z_i : i \in [n]\}$ be an IID sample drawn from $P_Z$ and let $\mathcal{B}$ be the uniform-mass binning scheme of size $B$ induced by $S$. There exists a universal constant $c' > 0$ such that for any $\delta \in (0,1)$, if $n \geq c' \cdot B \log(B/\delta)$, then $\Phi_{\mathrm{balance}}(\mathcal{B}, 2) = 1$ with probability at least $1 - \delta$.*

Lemma 3 states that

$$n \geq c' \cdot B \log\left(\frac{B}{\delta}\right) \qquad \Longrightarrow \qquad P\left[\mathcal{B} \text{ is 2-well-balanced with respect to } P_Z\right] \geq 1 - \delta.$$

While the value of the universal constant $c$ was not specified in the original reference [29], we remark that one may set, for example, $c' = 2420$, which can be verified by following their proof with $c'$ kept explicit.

The proof of Lemma 3 in [29] relies on a discretization argument that considers a fine-grained cover of $\mathcal{Z} = [0,1]$ consisting of disjoint intervals—namely, $\{I'_j : j \in [10B]\}$ such that $P[Z \in I'_j] = \frac{1}{10B}$ for all $j \in [10B]$—and then approximates each $I_b$ by a subset of the cover. As the authors of [29] remarked, this argument provides a tighter sample complexity upper bound than naïvely applying Chernoff bounds or a standard VC dimension argument, which would yield an upper bound of order $O\left(B^2 \log\left(\frac{B}{\delta}\right)\right)$. We omit the proof of Lemma 3 and refer interested readers to the referenced paper [29] for more details.

**Uniform concentration of bin-wise means.** Next, we argue that for the uniform-mass binning scheme $\mathcal{B}$ induced by an IID sample, the conditional empirical means of each bin concentrates to the population conditional expectation, uniformly for all bins in $\mathcal{B}$. Here we restate a result from [20].

**Lemma 4** ([20, Corollary 1]). *Let $P_Z$ be an absolutely continuous probability measure on $\mathcal{Z} = [0,1]$, and $S = \{Z_i : i \in [n]\}$ be an IID sample drawn from $P_Z$. Let $B \in \mathbb{N}$ such that $B \leq \frac{n}{2}$ and $\mathcal{B}$ be the uniform-mass binning scheme of size $B$ induced by $S$. Then for any $\delta \in (0,1)$,*

$$P\left[\Phi_{\mathrm{approx}}(\mathcal{B}; \varepsilon_\delta) = 1\right] \geq 1 - \delta \qquad \text{where} \qquad \varepsilon_\delta = \sqrt{\frac{1}{2(\lfloor n/B \rfloor - 1)} \log\left(\frac{2B}{\delta}\right)} + \frac{1}{\lfloor n/B \rfloor}. \quad (24)$$

Lemma 4 states that under the mild regularity condition of $P_Z$ being absolutely continuous, the uniform-mass binning accurately approximates all bin-wise conditional means as long as there are at least two samples per bin in the sense that

$$n \geq 2B \qquad \Longrightarrow \qquad P\left[\sup_{b \in [B]} |\hat{\mu}_b - \mu_b| \leq \sqrt{\frac{1}{2(\lfloor n/B \rfloor - 1)} \log\left(\frac{2B}{\delta}\right)} + \frac{1}{\lfloor n/B \rfloor}\right] \geq 1 - \delta.$$

## B.2 Conditional upper bounds on reliability risk and sharpness risk

In this section, we establish upper bounds on the reliability risk $R^{\mathrm{cal}}$ and the sharpness risk $R^{\mathrm{sha}}$ for $\hat{h}$ under the premise that $\Phi_{\mathrm{balance}}(\mathcal{B}; \alpha) = 1$ and $\Phi_{\mathrm{approx}}(\mathcal{B}; \varepsilon) = 1$ for appropriate parameters $\alpha, \varepsilon \in \mathbb{R}$.

**Preparation.** To avoid clutter in the lemma statements to follow, here we recall our problem setting and set several notation that will be used throughout this section. Recall that $P = P_{X,Y}$ is a joint distribution on $\mathcal{X} \times \mathcal{Y}$ and let $f : \mathcal{X} \to \mathcal{Z}$ is a measurable function. In addition, we let $\tilde{S} = \{(x_i, y_i) \in \mathcal{X} \times \mathcal{Y} : i \in [n]\}$ be an IID sample drawn from $P$, and let $S = \left\{(z, y) \in \mathcal{Z} \times \mathcal{Y} : (x,y) \in \tilde{S} \text{ and } z = f(x)\right\}$. Let $\mathcal{B}$ be the uniform-mass binning scheme induced by ($z$'s in) $S$, and let $\hat{h} = \hat{\mathcal{B}} : \mathcal{Z} \to \mathcal{Z}$ be the recalibration function derived from $\mathcal{B}$ as we described in Section 4.1; see (9). The dependence among $P, f, \tilde{S}, S, \mathcal{B}$, and $\hat{h}$ are summarized by a diagram in Figure 4.

Furthermore, we define the index function for a binning scheme to facilitate our analysis.

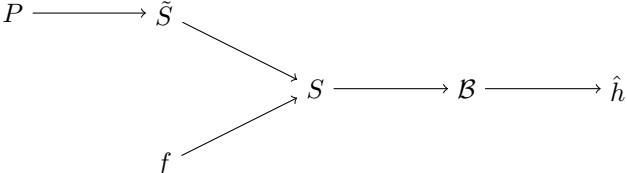

Figure 4: Stochastic dependence among $P, f, \tilde{S}, S, \mathcal{B}$, and $\hat{h}$.

**Definition 9.** *Let $\mathcal{B}$ be a binning scheme. The* index function *for $\mathcal{B}$ is the function $\beta : \mathcal{Z} \to [|\mathcal{B}|]$ such that*

$$\beta(z) = \sum_{I \in \mathcal{B}} \mathbb{1}_{(0, \sup I]}(z). \tag{25}$$

**Remark 6.** *Note that $\beta$ is a measurable function and defines an index function that identifies which bin of $\mathcal{B}$ the argument $z \in [0, 1]$ belongs to. Specifically, suppose that $\mathcal{B} = \{I_1, \ldots, I_B\}$ for some $B \in \mathbb{N}$ and there exists $u_0, u_1, \ldots, u_B \in [0, 1]$ such that (i) $0 = u_0 < u_1 < \cdots < u_B = 1$ and (ii) $I_b = (u_{b-1}, u_b]$ for all $b \in [B] \setminus \{1\}$ and $I_1 = [u_0, u_1]$. Then $\beta(z) = b$ if and only if $z \in I_b$.*

### B.2.1 Calibration risk upper bound

We observe that if a binning scheme $\mathcal{B}$ produces empirical means $\hat{\mu}_I$ that approximate the true means $\mu_I$ with error at most $\varepsilon$, then the calibration risk is upper bounded by $\varepsilon^2$.

**Lemma 5** (Calibration risk bound). *For any $\varepsilon \geq 0$, if $\Phi_{\mathrm{approx}}(\mathcal{B}; \varepsilon) = 1$, then*

$$R^{\mathrm{cal}}(\hat{h}; f, P) \leq \varepsilon^2.$$

*Proof of Lemma 5.* To begin with, we recall the definition of the calibration risk (Definition 3), and let $Z = f(X)$. Then we may write

$$
\begin{aligned}
R^{\mathrm{cal}}\big(\hat{h}; f, P\big) &= \mathbb{E}\left[\left(\hat{h}(Z) - \mathbb{E}[Y \mid \hat{h}(Z)]\right)^2\right] \\
&= \mathbb{E}\left[\mathbb{E}\left[\left(\hat{h}(Z) - \mathbb{E}[Y \mid \hat{h}(Z)]\right)^2 \Big| \beta(Z)\right]\right] && \because \text{the law of total expectation} \\
&= \mathbb{E}\left[\left(\hat{\mu}_{I_{\beta(Z)}} - \mu_{I_{\beta(Z)}}\right)^2\right] && \text{cf. (23)} \\
&\leq \max_{I \in \mathcal{B}} \left(\hat{\mu}_I - \mu_I\right)^2.
\end{aligned}
$$

Note that if $\Phi_{\mathrm{approx}}(\mathcal{B}; \varepsilon) = 1$, then $\max_{I \in \mathcal{B}} \left(\hat{\mu}_I - \mu_I\right)^2 \leq \varepsilon^2$. $\qquad \square$

We remark that the proof of Lemma 5 is a simple application of applying Hölder's inequality. Also, we note that a similar argument was considered in [20, Proposition 1] to establish the inequalities between the $L^p$-counterparts of the calibration risk, which they call the $\ell_p$-expected calibration error (ECE). In this work, we focus on the case $p = 2$.

### B.2.2 Sharpness risk upper bound

Next, we present an upper bound for the sharpness risk that diminishes as the binning scheme $\mathcal{B}$ becomes more balanced.

**Lemma 6** (Sharpness risk bound). *Suppose that the optimal post-hoc recalibration function $h^*_{f,P}$, cf. (7), is monotonically non-decreasing. Let $\alpha \in \mathbb{R}$ such that $\alpha \geq 1$. If $\Phi_{\mathrm{balance}}(\mathcal{B}, \alpha) = 1$, then*

$$R^{\mathrm{sha}}(\hat{h}; f, P) \leq \frac{\alpha}{|\mathcal{B}|}.$$

*Proof of Lemma 6.* Letting $Z = f(X)$, we can write the sharpness risk of $\hat{h}$ over $f$ with repsect to $P$ as

$$R^{\mathrm{sha}}(\hat{h}; f, P) := \mathbb{E}\left[\left(\mathbb{E}\big[Y \mid \hat{h}(Z)\big] - \mathbb{E}\left[Y \mid Z\right]\right)^2\right].$$

We recall the definition of the index function $\beta$ for $\mathcal{B}$ (Definition 9) and observe that

$$
\mathbb{E}\left[\left(\mathbb{E}\left[Y \mid \hat{h}(Z)\right] - \mathbb{E}\left[Y \mid Z\right]\right)^2\right]
$$

$$
\leq \mathbb{E}\left[\left|\mathbb{E}\left[Y \mid \hat{h}(Z)\right] - \mathbb{E}\left[Y \mid Z\right]\right|\right] \qquad \because \left|\mathbb{E}\left[Y \mid \hat{h}(Z)\right] - \mathbb{E}\left[Y \mid Z\right]\right| \leq 1
$$

$$
= \sum_{I \in \mathcal{B}} \mathbb{E}\left[\left|\mathbb{E}[Y \mid \hat{h}(Z)] - \mathbb{E}\left[Y \mid Z\right]\right| \cdot \mathbb{1}_I(Z)\right]
$$

$$
= \sum_{I \in \mathcal{B}} \mathbb{E}\left[\mathbb{E}\left[\left|\mathbb{E}[Y \mid \hat{h}(Z)] - \mathbb{E}\left[Y \mid Z\right]\right| \cdot \mathbb{1}_I(Z) \,\middle|\, \beta(Z)\right]\right] \qquad \because \text{the law of total expectation}
$$

$$
= \sum_{I \in \mathcal{B}} P[Z \in I] \cdot \mathbb{E}\left[\mathbb{E}\left[\left|\mathbb{E}[Y \mid \hat{h}(Z)] - \mathbb{E}\left[Y \mid Z\right]\right| \,\middle|\, Z \in I\right]\right] \qquad \because \text{Remark 6}
$$

$$
\leq \sum_{I \in \mathcal{B}} P[Z \in I] \cdot \left(\sup_{z \in I} h_{f,P}^*(z) - \inf_{z \in I} h_{f,P}^*(z)\right) \qquad \because \text{by definition of } h_{f,P}^*;\ \text{cf. (7)}
$$

$$
\leq \sum_{I \in \mathcal{B}} \frac{\alpha}{|\mathcal{B}|} \cdot \left(\sup_{z \in I} h_{f,P}^*(z) - \inf_{z \in I} h_{f,P}^*(z)\right) \qquad \because \Phi_{\text{balance}}(\mathcal{B}, \alpha) = 1
$$

$$
\leq \frac{\alpha}{|\mathcal{B}|}.
$$

The inequality in the last line follows from the facts that (i) $I \in \mathcal{B}$ are mutually exclusive and (ii) $h_{f,P}^*(z) \in [0,1]$ and $h_{f,P}^*$ is monotone non-decreasing. $\qquad\square$

Our proof of Lemma 6 relies on similar techniques that are used in [29, Lemmas D.5 and D.6]. However, we note that we obtain an improved constant — 1 as opposed to 2 in [29, Lemma D.6] — with a more refined analysis.

**An improved rate with additional assumptions.** It is possible to improve the rate of the sharpness risk upper bound from $O(|\mathcal{B}|^{-1})$ to $O(|\mathcal{B}|^{-2})$ with an additional regularity assumption on $h_{f,\mathcal{B}}^*$.

Recall that we assumed in (A3) that there exists $K > 0$ such that if $z_1 \leq z_2$, then $h_{f,P}^*(z_2) - h_{f,P}^*(z_1) \leq K \cdot \left(F_Z(z_2) - F_Z(z_1)\right)$, that is, $h_{f,P}^*$ is $K$-smooth with respect to $F_Z$. This posits that the conditional probability $P[Y = 1|Z = z]$ of the target variable $Y$ given a forecast variable $Z$ cannot vary too much in regions where the density of $Z$ is low, or where the forecast is rarely issued. This is a reasonable assumption because if $P[Y = 1|Z]$ changes too rapidly with respect to $Z$, then it suggests that we need additional information about $Y$ beyond what $Z$ can provide in order to improve the quality of forecasts. We remark that (A3) is indeed a fairly mild assumption to impose on, however, is not a trivial one.

**Remark 7** (Mildness of (A3)). *Suppose that $Z = f(X)$ has a density $p_Z$ that is uniformly lower bounded by $\epsilon$ on the support of $Z$. If $h_{f,P}^*$ is $L$-Lipschitz, then $h_{f,P}^*$ is $(L/\epsilon)$-smooth with respect to $F_Z$. This also provides a sufficient condition to verify (A3) in practice.*

**Remark 8** (Non-triviality of (A3)). *Notice that even if $F_Z$ is absolutely continuous and $h_{f,P}^*$ is continuous, the smoothness constant $K$ could become large if the prediction $Z$ is heavily miscalibrated. For instance, in Figure 6, $h_{f,P}^*(z)$ is changing fast in the interval $[0.5, 0.75]$ where $p_Z(z)$ is small, which results in a larger value of $K$ that can even diverge if $p_Z(z) \to 0$.*

Here we define the notion of $\psi$-smoothness to formalize Assumption (A3), and then present an improved upper bound for the sharpness risk.

**Definition 10** ($\psi$-smoothness). *Let $K \in \mathbb{R}_+$ and $\psi : [0,1] \to [0,1]$ be a monotone non-decreasing function. A function $\phi : [0,1] \to [0,1]$ is $K$-smooth with respect to $\psi$ if for any $z_1, z_2 \in [0,1]$ such that $z_1 \leq z_2$,*

$$
\left|\phi(z_2) - \phi(z_1)\right| \leq K \cdot \left(\psi(z_2) - \psi(z_1)\right). \tag{26}
$$

**Lemma 7** (Improved sharpness risk bound). *Suppose that the function $h_{f,P}^*(z)$ defined in (7) is monotonically non-decreasing and $K$-smooth with respect to $F_Z$ for some $K \geq 0$, where $F_Z$ is the cumulative distribution function of $Z = f(X)$. If $\Phi_{\text{balance}}(\mathcal{B}, \alpha) = 1$, then*

$$
R^{\text{sha}} \leq \frac{K^2 \alpha^3}{B^2}.
$$

*Proof of Lemma 7.* Let $Z = f(X)$ and $B = |\mathcal{B}|$. For each $b \in [B]$, we let $z_{b,\mathrm{max}} := \sup I_b$ and $z_{b,\mathrm{min}} := \inf I_b$. Then we have

$$R^{\mathrm{sha}}(\hat{h}; f, P)$$

$$= \mathbb{E}\left[\left(\mathbb{E}[Y \mid \hat{h}(Z)] - \mathbb{E}[Y \mid Z]\right)^2\right]$$

$$= \mathbb{E}\left[\mathbb{E}\left[\left(\mathbb{E}[Y \mid \hat{h}(Z)] - \mathbb{E}[Y \mid Z]\right)^2 \,\Big|\, \beta(Z)\right]\right]$$

$$= \sum_{b=1}^{B} P[Z \in I_b] \cdot \mathbb{E}\left[\left(\mathbb{E}[Y \mid \hat{h}(Z)] - \mathbb{E}[Y \mid Z]\right)^2 \,\Big|\, \beta(Z) = b\right]$$

$$\leq \sum_{b=1}^{B} P[Z \in I_b] \cdot \left(h_{f,P}^*(z_{b,\mathrm{max}}) - h_{f,P}^*(z_{b,\mathrm{min}})\right)^2 \qquad \because h_{f,P}^* \text{ is non-decreasing}$$

$$\leq \sum_{b=1}^{B} P[Z \in I_b] \cdot \left(K \cdot \left(F_Z(z_{b,\mathrm{max}}) - F_Z(z_{b,\mathrm{min}})\right)\right)^2 \qquad \because h_{f,P}^* \text{ is K-smooth w.r.t. } F_Z$$

$$= \sum_{b=1}^{B} K^2 \cdot P[Z \in I_b]^3$$

$$\leq K^2 \sum_{b=1}^{B} \left(\frac{\alpha}{B}\right)^3 \qquad\qquad\qquad \because \Phi_{\mathrm{balance}}(\mathcal{B}, \alpha) = 1$$

$$= \frac{K^2 \alpha^3}{B^2}.$$

$\square$

**Remark 9** (Tightness of the rate $O(B^{-2})$). *The asymptotic rate $R^{\mathrm{sha}} = O(B^{-2})$ is tight and cannot be further improved without additional assumptions. For instance, let's consider a uniform-mass binning of size $B$ on $Z \sim Uniform[0, 1]$. In the population limit, each bin has width $1/B$ and within-bin variance $1/(12B^2)$. Thus, the sharpness risk, obtained by taking expectation of the conditional variance (per each bin), is $1/(12B^2)$, attaining the rate $B^{-2}$.*

## B.3   Completing the proof of Theorem 1

*Proof of Theorem 1.* For given $\delta \in (0, 1)$, let $\delta_1 = \delta_2 = \delta/2$. Then we observe that

$$n \geq c' \cdot |\mathcal{B}| \log\left(\frac{|\mathcal{B}|}{\delta_1}\right) \qquad \Longrightarrow \qquad P[\Phi_{\mathrm{balance}}(\mathcal{B}, 2) = 1] \geq 1 - \delta_1 \qquad \text{by Lemma 3}$$

$$n \geq 2|\mathcal{B}| \qquad \Longrightarrow \qquad P[\Phi_{\mathrm{approx}}(\mathcal{B}, \varepsilon_{\delta_2}) = 1] \geq 1 - \delta_2 \qquad \text{by Lemma 4}$$

where $c' > 0$ is the universal constant that appears in Lemma 3 and

$$\varepsilon_{\delta_2} = \sqrt{\frac{1}{2(\lfloor n/|\mathcal{B}| \rfloor - 1)} \log\left(\frac{2|\mathcal{B}|}{\delta_2}\right)} + \frac{1}{\lfloor n/|\mathcal{B}| \rfloor}.$$

Observe that $\delta_1 = \frac{\delta}{2} < \frac{1}{2}$ and $|\mathcal{B}| \geq 1$, and thus, $\log\left(\frac{|\mathcal{B}|}{\delta_1}\right) \geq \log 2$. Letting $c := \max\{c', \frac{2}{\log 2}\}$ and applying the union bound, we have

$$n \geq c \cdot |\mathcal{B}| \log\left(\frac{2|\mathcal{B}|}{\delta}\right) \qquad \Longrightarrow \qquad P[\Phi_{\mathrm{balance}}(\mathcal{B}, 2) = 1 \text{ and } \Phi_{\mathrm{approx}}(\mathcal{B}, \varepsilon_{\delta/2}) = 1] \geq 1 - \delta.$$

Next, we observe that if $\Phi_{\mathrm{balance}}(\mathcal{B}, 2) = 1$ and $\Phi_{\mathrm{approx}}(\mathcal{B}, \varepsilon_{\delta_2}) = 1$, then

$$R^{\mathrm{cal}}(\hat{h}; f; P) \leq (\varepsilon_{\delta_2})^2 \qquad\qquad \text{by Lemma 5,}$$

$$R^{\mathrm{sha}}(\hat{h}; f, P) \leq \frac{2}{|\mathcal{B}|}, \qquad\qquad \text{by Lemma 6.}$$

Additionally, if the Assumption (A3) also holds, then we obtain a stronger upper bound on $R^{\mathrm{sha}}(\hat{h}; f, P)$ by Lemma 7:

$$R^{\mathrm{sha}}(\hat{h}; f, P) \leq \frac{8K^2}{|\mathcal{B}|^2}.$$

$\square$

# C    Proof of Theorem 2

This section contains a proof of Theorem 2. Prior to the proof, in Section C.1, we provide several lemmas that will be useful in our proof. Thereafter, we present a proof of Theorem 2 in its entirety in Section C.2.

## C.1    Useful lemmas

### C.1.1    Concentration of $\hat{w}_k$ to $w_k^*$

First of all, we recall the binomial Chernoff bound, which is a classical result about the concentration of measures that can be found in standard textbooks on probability theory.

**Lemma 8** (Binomial Chernoff bound). *Let $X_i$ be IID Bernoulli random variables with parameters $p \in (0, 1)$, and let $S_n := \frac{1}{n} \sum_{i=1}^{n} X_i$. Then for any $\delta \in \mathbb{R}$ such that $0 < \varepsilon < 1$,*

$$P\left[S_n \geq (1 + \varepsilon)p\right] \leq \exp\left(-\frac{\varepsilon^2 p}{3} n\right),$$

$$P\left[S_n \leq (1 - \varepsilon)p\right] \leq \exp\left(-\frac{\varepsilon^2 p}{2} n\right).$$

It follows from Lemma 8 that for any $\varepsilon, \delta \in (0, 1)$,

$$n \geq \frac{3}{\varepsilon^2 p} \log\left(\frac{2}{\delta}\right) \qquad \implies \qquad P\left(\frac{|S_n - p|}{p} > \varepsilon\right) \leq \delta. \tag{27}$$

Let $P, Q$ be two distributions on $\mathcal{Y} = \{0, 1\}$, and let $\mathcal{D}_P \sim P$, $\mathcal{D}_Q \sim Q$ denote IID samples of size $n_P$, $n_Q$, respectively. Recall from (13) and (16) that for each $k \in \{0, 1\}$, we define

$$w_k^* = \frac{P_Q[Y = k]}{P_P[Y = k]}, \qquad \text{and} \qquad \hat{w}_k = \frac{P_{\mathcal{D}_Q}[Y = k]}{P_{\mathcal{D}_P}[Y = k]}.$$

Then, we let

$$\rho_0 := \frac{\hat{w}_0}{w_0^*} \qquad \text{and} \qquad \rho_1 := \frac{\hat{w}_1}{w_1^*}. \tag{28}$$

Now we define another parameterized family of Boolean-valued functions $\Phi_{\mathrm{ratio}}(\mathcal{D}_P, \mathcal{D}_Q; \beta)$ as follows. Given $\mathcal{D}_P \sim P$, $\mathcal{D}_Q \sim Q$, and $\beta \in \mathbb{R}$ such that $1 < \beta \leq 2$,

$$\Phi_{\mathrm{ratio}}(\mathcal{D}_P, \mathcal{D}_Q; \beta) := \mathbb{1}\left\{\frac{1}{\beta} \leq \rho_k \leq \beta, \ \forall k \in \{0, 1\}\right\}. \tag{29}$$

**Corollary 9.** *Let $P, Q$ be two distributions on $\mathcal{Y} = \{0, 1\}$, and let $\mathcal{D}_P \sim P$, $\mathcal{D}_Q \sim Q$ denote IID samples of size $n_P$, $n_Q$, respectively. For each $k \in \{0, 1\}$, let $p_k := P_P[Y = k]$ and $q_k := P_Q[Y = k]$. Likewise, we let $\hat{p}_k = \frac{1}{n_P} \sum_{y_i \in \mathcal{D}_P} \mathbb{1}\{y_i = k\}$ and $\hat{q}_k = \frac{1}{n_Q} \sum_{y_i \in \mathcal{D}_Q} \mathbb{1}\{y_i = k\}$. For any $\delta \in (0, 1)$ and any $\beta \in (1, 2]$, if*

$$n_P \geq \frac{27}{(\beta - 1)^2 \min\{p_0, p_1\}} \log\left(\frac{8}{\delta}\right) \quad \text{and} \quad n_Q \geq \frac{27}{(\beta - 1)^2 \min\{q_0, q_1\}} \log\left(\frac{8}{\delta}\right),$$

*then*

$$P\left(\Phi_{\mathrm{ratio}}(\mathcal{D}_P, \mathcal{D}_Q; \beta) = 1\right) \geq 1 - \delta.$$

*Proof of Corollary 9.* Let $\varepsilon = \frac{\beta-1}{3}$. Since $\frac{1+x}{1-x} \leq 1 + 3x$ for all $x \in [0, 1/3]$, we have $\frac{1}{\beta} \leq \frac{1-\varepsilon}{1+\varepsilon} < \frac{1+\varepsilon}{1-\varepsilon} \leq \beta$. Then it follows from (27) that for each $k \in \{0, 1\}$,

$$n_P \geq \frac{3}{\varepsilon^2 p_k} \log\left(\frac{8}{\delta}\right) \qquad \Longrightarrow \qquad P\left(\frac{|\hat{p}_k - p_k|}{p_k} > \varepsilon\right) \leq \frac{\delta}{4},$$

$$n_Q \geq \frac{3}{\varepsilon^2 q_k} \log\left(\frac{8}{\delta}\right) \qquad \Longrightarrow \qquad P\left(\frac{|\hat{q}_k - q_k|}{q_k} > \varepsilon\right) \leq \frac{\delta}{4}.$$

Applying the union bound, we obtain the following implication:

$$n_P \geq \frac{3}{\varepsilon^2 \min\{p_0, p_1\}} \log\left(\frac{8}{\delta}\right) \text{ and } n_Q \geq \frac{3}{\varepsilon^2 \min\{q_0, q_1\}} \log\left(\frac{8}{\delta}\right)$$

$$\Longrightarrow \qquad P\left(\max_{k \in \{0,1\}} \frac{|\hat{p}_k - p_k|}{p_k} > \varepsilon \ \text{ or } \max_{k \in \{0,1\}} \frac{|\hat{q}_k - q_k|}{q_k} > \varepsilon\right) \leq \delta$$

$$\Longrightarrow \qquad P\left(\max_{k \in \{0,1\}} \rho_k > \frac{1+\varepsilon}{1-\varepsilon} \ \text{ or } \min_{k \in \{0,1\}} \rho_k < \frac{1-\varepsilon}{1+\varepsilon}\right) \leq \delta$$

$$\Longrightarrow \qquad P\left(\max_{k \in \{0,1\}} \rho_k > \beta \ \text{ or } \min_{k \in \{0,1\}} \rho_k < \frac{1}{\beta}\right) \leq \delta.$$

$\square$

### C.1.2 Regularity of the Shift Correction Function

**Lemma 10.** *Let $w = (w_0, w_1) \in \mathbb{R}^2$ such that $w_0, w_1 > 0$ and $w_0 + w_1 = 1$. The function $g_w : [0, 1] \to [0, 1]$ such that $g_w(z) = \frac{w_1 z}{w_1 z + w_0 (1-z)}$ is $L$-Lipschitz where $L = \max\left\{\frac{w_1}{w_0}, \frac{w_0}{w_1}\right\}$.*

*Proof of Lemma 10.* First of all, consider the first-order derivative of $g_w$:

$$\frac{d}{dz} g_w(z) = \frac{w_1 \cdot [w_1 z + w_0(1-z)] - w_1 z \cdot (w_1 - w_0)}{[w_1 z + w_0(1-z)]^2} = \frac{w_1 w_0}{[w_1 z + w_0(1-z)]^2}.$$

We observe that $g_w$ is monotone increasing as $\frac{d}{dz} g_w(z) > 0$ for all $z \in [0, 1]$. Next, we consider the second-order derivative of $g_w$:

$$\frac{d^2}{dz^2} g_w(z) = \frac{2 w_0 w_1 \cdot (w_0 - w_1)}{[w_1 z + w_0(1-z)]^3} \begin{cases} > 0, & \forall z \in [0, 1] \quad \text{if } w_0 > w_1, \\ = 0, & \forall z \in [0, 1] \quad \text{if } w_0 = w_1, \\ < 0, & \forall z \in [0, 1] \quad \text{if } w_0 < w_1. \end{cases}$$

Therefore,

$$\sup_{z \in [0,1]} \frac{d}{dz} g_w(z) = \begin{cases} \left.\frac{d}{dz} g_w(z)\right|_{z=1} = \frac{w_0}{w_1} & \text{if } w_0 > w_1, \\ \left.\frac{d}{dz} g_w(z)\right|_{z=0} = \frac{w_1}{w_0} & \text{if } w_0 \leq w_1. \end{cases}$$

$\square$

**Lemma 11.** *Let $P, Q$ be joint distributions of $(X, Y) \in \mathcal{X} \times \{0, 1\}$, and let $w_k = \frac{P[Y=k]}{Q[Y=k]}$ for $k \in \{0, 1\}$. If $P, Q$ satisfy the label shift assumption (Definition 7), i.e., if Assumptions (B1) and (B2) hold, then for any measurable function $f : \mathcal{X} \to \mathbb{R}$, the following two-sided inequality holds:*

$$\min_{k \in \{0,1\}} w_k \leq \frac{\mathbb{E}_Q[f(X)]}{\mathbb{E}_P[f(X)]} \leq \max_{k \in \{0,1\}} w_k. \tag{30}$$

*Proof of Lemma 11.* First of all, we observe that

$\mathbb{E}_Q[f(X)] = \mathbb{E}_Q\big[\mathbb{E}_Q[f(X) \mid Y]\big]$     by the law of total expectation

$$= \sum_{k=0}^{1} P_Q[Y=k] \cdot \mathbb{E}_Q\big[f(X) \mid Y\big]$$

$$= \sum_{k=0}^{1} \big(w_k \cdot P_P[Y=k]\big) \cdot \mathbb{E}_P\big[f(X) \mid Y\big]. \quad \text{by definition of } w_k \ \& \text{ the label shift assumption}$$

Thus, it follows that $\min_k w_k \cdot \mathbb{E}_P[f(X)] \leq \mathbb{E}_Q[f(X)] \leq \max_k w_k \cdot \mathbb{E}_P[f(X)]$. $\square$

## C.2 Completing the proof of Theorem 2

*Proof of Theorem 2.* This proof is presented in four steps. In Step 1, we establish a simple upper bound for the risk $R_Q(\hat{h}_Q; f)$ that consists of two error terms: the first term quantifies the error introduced by the estimated label shift correction, $\hat{g}$, while the second term quantifies the error due to the estimated recalibration function, $\hat{h}_P$. In Steps 2 and 3, we derive separate upper bounds for these two error terms. Finally, in Step 4, we combine the results from Steps 1-3 to obtain a comprehensive upper bound for $R_Q$, which concludes the proof.

**Step 1. Decomposition of $R_Q$.** Recalling the definition of the risk $R_Q$, cf. (5), we obtain the following inequality:

$$R_Q(\hat{h}_Q; f) = \mathbb{E}_Q\left[\left(\hat{h}_Q \circ f(X) - \mathbb{E}_Q[Y|f(X)]\right)^2\right]$$

$$= \mathbb{E}_Q\left[\left(\hat{g} \circ \hat{h}_P \circ f(X) - g^* \circ \hat{h}_P \circ f(X) + g^* \circ \hat{h}_P \circ f(X) - \mathbb{E}_Q[Y|f(X)]\right)^2\right]$$

$$\overset{(a)}{\le} 2 \cdot \left\{ \underbrace{\mathbb{E}_Q\left[\left(\hat{g} \circ \hat{h}_P \circ f(X) - g^* \circ \hat{h}_P \circ f(X)\right)^2\right]}_{=:T_1} \right. \tag{31}$$

$$\left. + \underbrace{\mathbb{E}_Q\left[\left(g^* \circ \hat{h}_P \circ f(X) - \mathbb{E}_Q[Y|f(X)]\right)^2\right]}_{=:T_2} \right\}, \tag{32}$$

where (a) follows from the simple inequality $(a+b)^2 \le 2(a^2+b^2)$ for all $a, b \in \mathbb{R}$.

In Step 2 and Step 3 of this proof, we establish separate upper bounds for the two terms, $T_1, T_2$.

**Step 2. An upper bound for $T_1$.** Recall from (13) and (16) that

$$g^*(z) = \frac{w_1^* z}{w_1^* z + w_0^*(1-z)} \qquad \text{where} \qquad w_k^* = \frac{Q[Y=k]}{P[Y=k]}, \quad \forall k \in \{0, 1\},$$

$$\hat{g}(z) = \frac{\hat{w}_1 z}{\hat{w}_1 z + \hat{w}_0(1-z)} \qquad \text{where} \qquad \hat{w}_k = \frac{\hat{Q}[Y=k]}{\hat{P}[Y=k]}, \quad \forall k \in \{0, 1\}.$$

Let

$$\rho_0 := \frac{\hat{w}_0}{w_0^*} \qquad \text{and} \qquad \rho_1 := \frac{\hat{w}_1}{w_1^*}. \tag{33}$$

Then we observe that for any $z \in (0, 1)$,

$$|\hat{g}(z) - g^*(z)| = \left| \frac{\hat{w}_1 z}{\hat{w}_1 z + \hat{w}_0(1-z)} - \frac{w_1^* z}{w_1^* z + w_0^*(1-z)} \right|$$

$$= \left| \frac{(\hat{w}_1 w_0^* - w_1^* \hat{w}_0) \cdot z(1-z)}{[\hat{w}_1 z + \hat{w}_0(1-z)] \cdot [w_1^* z + w_0^*(1-z)]} \right|$$

$$\le \left| \frac{(\hat{w}_1 w_0^* - w_1^* \hat{w}_0) \cdot z(1-z)}{(\hat{w}_1 w_0^* + w_1^* \hat{w}_0) \cdot z(1-z)} \right|$$

$$= \left| \frac{\hat{w}_1 w_0^* - w_1^* \hat{w}_0}{\hat{w}_1 w_0^* + w_1^* \hat{w}_0} \right|$$

$$= \frac{|\rho_0 - \rho_1|}{\rho_0 + \rho_1}.$$

Moreover, $\hat{g}(0) = g^*(0) = 0$ and $\hat{g}(1) = g^*(1) = 1$. Letting $Z_{\hat{h}} := \hat{h}_P \circ f(X)$, we obtain

$$T_1 = \mathbb{E}_Q\left[\left(\hat{g}(Z_{\hat{h}}) - g^*(Z_{\hat{h}})\right)^2\right] \le \left(\frac{\rho_0 - \rho_1}{\rho_0 + \rho_1}\right)^2. \tag{34}$$

It remains to establish probabilistic tail bounds for $\rho_0, \rho_1$, which we will accomplish in Step 4 of this proof.

**Step 3. An upper bound for $T_2$.**  We observe that

$$
\begin{aligned}
T_2 &= \mathbb{E}_Q\left[\left(g^* \circ \hat{h}_P \circ f(X) - \mathbb{E}_Q[Y \mid f(X)]\right)^2\right] \\
&= \mathbb{E}_Q\left[\left(g^* \circ \hat{h}_P \circ f(X) - g^*\left(\mathbb{E}_P[Y \mid f(X)]\right)\right)^2\right] && \because \text{Label shift assumption, cf. (14)} \\
&\leq \left(\frac{w_{\max}^*}{w_{\min}^*}\right)^2 \cdot \mathbb{E}_Q\left[\left(\hat{h}_P \circ f(X) - \mathbb{E}_P[Y \mid f(X)]\right)^2\right] && \because g^* \text{ is } \frac{w_{\max}^*}{w_{\min}^*}\text{-Lipschitz, cf. Lemma 10} \\
&\leq \left(\frac{w_{\max}^*}{w_{\min}^*}\right)^2 \cdot w_{\max}^* \cdot \mathbb{E}_P\left[\left(\hat{h}_P \circ f(X) - \mathbb{E}_P[Y \mid f(X)]\right)^2\right] && \because \text{by Lemma 11} \\
&= \frac{w_{\max}^*{}^3}{w_{\min}^*{}^2} \cdot R_P\left(\hat{h}_P; f\right).
\end{aligned}
$$

**Step 4. Concluding the proof.**  For given $\delta \in (0,1)$, let[2] $\delta_1 = \delta_2 = \delta/4$ and $\delta_3 = \delta/2$. We observe that

$$
\begin{aligned}
n_P &\geq c' \cdot |\mathcal{B}| \log\left(\frac{|\mathcal{B}|}{\delta_1}\right) && \Longrightarrow && P\left[\Phi_{\text{balance}}(\mathcal{B}, 2) = 1\right] \geq 1 - \delta_1 && \text{by Lemma 3} \\
n_P &\geq 2|\mathcal{B}| && \Longrightarrow && P\left[\Phi_{\text{approx}}(\mathcal{B}, \varepsilon_{\delta_2}) = 1\right] \geq 1 - \delta_2 && \text{by Lemma 4}
\end{aligned}
$$

where $c' > 0$ is the universal constant that appears in Lemma 3 and

$$
\varepsilon_{\delta_2} = \sqrt{\frac{1}{2(\lfloor n/|\mathcal{B}| \rfloor - 1)} \log\left(\frac{2|\mathcal{B}|}{\delta_2}\right)} + \frac{1}{\lfloor n/|\mathcal{B}| \rfloor}.
$$

Furthermore, assuming

$$
n_P \geq \frac{27}{\min\{p_0, p_1\}} \log\left(\frac{8}{\delta_3}\right) \qquad \text{and} \qquad n_Q \geq \frac{27}{\min\{q_0, q_1\}} \log\left(\frac{8}{\delta_3}\right),
$$

we may define $\beta_{\delta_3}$ as a function of $n_P, n_Q$ and $\delta_3$ such that

$$
\beta_{\delta_3} = \beta_{\delta_3}(n_P, n_Q) := 1 + \sqrt{\max\left\{\frac{1}{n_P \cdot \min\{p_0, p_1\}}, \frac{1}{n_Q \cdot \min\{q_0, q_1\}}\right\} \cdot 27 \log\left(\frac{8}{\delta_3}\right)}. \quad (35)
$$

Then it follows from Corollary 9 that

$$
P\left(\Phi_{\text{ratio}}(\mathcal{D}_P, \mathcal{D}_Q; \beta_0) = 1\right) \geq 1 - \delta_3.
$$

Observe that $\delta_1 = \frac{\delta}{4} < \frac{1}{4}$ and $|\mathcal{B}| \geq 4$, and thus, $\log\left(\frac{|\mathcal{B}|}{\delta_1}\right) \geq \log 16 \geq 2$. Let $c = c'$. Since $c' \geq 1$ and $\log\left(\frac{|\mathcal{B}|}{\delta_1}\right) \geq \log\left(\frac{16}{\delta}\right) = \log\left(\frac{8}{\delta_3}\right)$, we notice that

$$
\begin{aligned}
n_P &\geq \max\left\{c, \frac{27}{\min\{p_0, p_1\}}\right\} \cdot |\mathcal{B}| \log\left(\frac{4|\mathcal{B}|}{\delta}\right) \\
&\Longrightarrow \quad n_P \geq \max\left\{c' \cdot |\mathcal{B}| \log\left(\frac{|\mathcal{B}|}{\delta_1}\right), \, 2|\mathcal{B}|, \, \frac{27}{\min\{p_0, p_1\}} \log\left(\frac{8}{\delta_3}\right)\right\}.
\end{aligned}
$$

In summary, we obtain that for any given $\delta \in (0,1)$,

$$
\begin{aligned}
n_P &\geq \max\left\{c, \frac{27}{\min\{p_0, p_1\}}\right\} \cdot |\mathcal{B}| \log\left(\frac{4|\mathcal{B}|}{\delta}\right) \quad \text{and} \quad n_Q \geq \frac{27}{\min\{q_0, q_1\}} \log\left(\frac{16}{\delta}\right) \\
&\Longrightarrow \quad P\left[\Phi_{\text{balance}}(\mathcal{B}, 2) = 1 \ \& \ \Phi_{\text{approx}}(\mathcal{B}, \varepsilon_{\delta/4}) = 1 \ \& \ \Phi_{\text{ratio}}(\mathcal{D}_P, \mathcal{D}_Q; \beta_{\delta/2}) = 1\right] \geq 1 - \delta.
\end{aligned}
$$
$$(36)$$

---

[2]We remark that our decomposition of $\delta$ into $\delta_1, \delta_2, \delta_3$ is arbitrary, and is intended to simplify the subsequent analysis.

Conditioned on the event $\Phi_{\text{balance}}(\mathcal{B}, 2) = 1$ & $\Phi_{\text{approx}}(\mathcal{B}, \varepsilon_{\delta/4}) = 1$ & $\Phi_{\text{ratio}}(\mathcal{D}_P, \mathcal{D}_Q; \beta_{\delta/2}) = 1$,

$$T_1 \leq \left(\frac{|\rho_0 - \rho_1|}{\rho_0 + \rho_1}\right)^2 \leq \left(\frac{\beta_{\delta/2} - \frac{1}{\beta_{\delta/2}}}{\beta_{\delta/2} + \frac{1}{\beta_{\delta/2}}}\right)^2 \leq \left(\beta_{\delta/2} - 1\right)^2, \quad \because (34); \text{ also, see } (29)$$

$$T_2 \leq \frac{w_{\max}^{*3}}{w_{\min}^{*2}} \cdot R_P(\hat{h}_P; f)$$

$$\leq \frac{w_{\max}^{*3}}{w_{\min}^{*2}} \cdot \left(\varepsilon_{\delta/4}^2 + \frac{2}{|\mathcal{B}|}\right). \qquad\qquad \because \text{ proof of Theorem 1; Lemmas 5 & 6}$$

Note that if Assumption (A3) holds, then we additionally have

$$T_2 \leq \frac{w_{\max}^{*3}}{w_{\min}^{*2}} \cdot \left(\varepsilon_{\delta/4}^2 + \frac{8K^2}{|\mathcal{B}|^2}\right).$$

Inserting these upper bounds for $T_1$ and $T_2$ into (31), (32) and recalling the expression for $\beta$ in (35), we complete the proof. $\qquad\square$

## D  Proof sketch of the argument in Remark 5

Recall our composite recalibration function,

$$\hat{h}_Q = \hat{g} \circ \hat{h}_P, \tag{37}$$

does not use the features in $\mathcal{D}_Q$. Specifically, $\hat{g}$ is parameterized by $\hat{w} = (\hat{w}_0, \hat{w}_1)$, which can be estimated using only labels in $\mathcal{D}_P$ and $\mathcal{D}_Q$, cf. (16). According to Theorem 2, $R_Q(\hat{h}_Q) = O(n_Q^{-1})$ with high probability for sufficiently large $n_P$.

Now suppose we are given an unlabeled target sample with unknown label shift. We can estimate $w$ using the target features via a maximum likelihood label shift estimation approach [14], yielding $\hat{w}^{\text{ML}} = (\hat{w}_0^{\text{ML}}, \hat{w}_1^{\text{ML}})$. and the calibrated classifier $\hat{h} \circ f$. This results in a different composite recalibration function than Equation (37),

$$\hat{h}_Q^{\text{ML}} = \hat{g}^{\text{ML}} \circ \hat{h}_P, \tag{38}$$

where $\hat{g}^{\text{ML}} : [0,1] \to [0,1]$ is defined as $\hat{g}^{\text{ML}}(z) = \hat{w}_1^{\text{ML}} z / (\hat{w}_1^{\text{ML}} z + \hat{w}_0^{\text{ML}}(1-z))$. We claim in Remark 5 that, for sufficiently large $n_P$, the composite recalibration function in Equation (38) achieves $R_Q(\hat{h}_Q^{\text{ML}}) = O(n_Q^{-1})$ with high probability, enjoying the same convergence rate as $\hat{h}_Q$ (Equation 37). Here we give a proof sketch.

*Proof sketch.* Suppose we use the maximum likelihood approach in [14] to estimate $w$. We want to show $R_Q(\hat{h}_Q^{\text{ML}}) = O(n_Q^{-1})$ with high probability for sufficiently large $n_P$. Recall $R_Q(\hat{h}_Q) \leq 2(T_1 + T_2)$ according to Equation (31) and (32), and label shift estimation error only affects $T_1$, so it is sufficient to show $T_1 = O(n_Q^{-1})$ with high probability.

For sufficiently large $n_P$, $\hat{h}_P \circ f$ is sufficient calibrated, so $\|\hat{w} - w\|_2^2 = O(n_Q^{-1})$ by Theorem 3 in [14]. Since

$$\|\hat{w} - w\|_2^2 = \sum_{k \in \{0,1\}} (\rho_k - 1)^2 w_k^2 \geq w_{\min}^{*2} \sum_{k \in \{0,1\}} (\rho_k - 1)^2 \geq w_{\min}^{*2} \max_{k \in \{0,1\}} (\rho_k - 1)^2, \tag{39}$$

we have $\rho_k \in [1-\alpha, 1+\alpha]$ for $k \in \{0, 1\}$, where $\alpha = \frac{\|\hat{w} - w\|_2}{w_{\min}^*} > 0$. For sufficiently small $\|\hat{w} - w\|_2^2$, we can control $\alpha < 0.5$, which bounds $T_1$ in (34):

$$T_1 \leq \left(\frac{\rho_0 - \rho_1}{\rho_0 + \rho_1}\right)^2 \leq \frac{2\alpha}{2 - 2\alpha} \leq 2\alpha = 2\frac{\|\hat{w} - w\|_2}{w_{\min}^*} = O(n_Q^{-1}). \tag{40}$$

The rest of the proof are the same with Appendix C.2. $\qquad\square$

# E    Details on the experiments

In Section E.1 and E.3, we consider a family of joint distributions $\mathcal{D}(\pi)$ of $X$ and $Y$, where $Y \sim \text{Bernoulli}(\pi)$, $X \mid Y = 0 \sim N(-2, 1)$, and $X \mid Y = 1 \sim N(2, 1)$. Suppose we are given $f(x) = \sigma(x) := 1/(1 + e^{-x})$, for $x \in \mathbb{R}$, as a probabilistic classifier. The optimal recalibration function can be derived as

$$h_{f,P}^*(z) = P[Y = 1 \mid f(X) = z] = \sigma(4\sigma^{-1}(z)). \tag{41}$$

In Section E.2, we consider a parametric family of recalibration functions called beta calibration [28]: $\mathcal{H}_{\text{beta}} = \{h_{\text{beta}}(\cdot; a, b, c) : a \geq 0, b \geq 0, c \in \mathbb{R}\}$, where $h_{\text{beta}}(\cdot; a, b, c) : [0, 1] \to [0, 1]$ is defined as

$$h_{\text{beta}}(z; a, b, c) = \frac{1}{1 + 1/\left(e^c \frac{z^a}{(1-z)^b}\right)}. \tag{42}$$

In addition, consider a subfamily $\mathcal{H}_{\text{logit-normal}} \subset \mathcal{H}_{\text{beta}}$ defined as $\mathcal{H}_{\text{logit-normal}} = \{h_{\text{logit-normal}}(\cdot; a, c) := h_{\text{beta}}(\cdot; a, a, c) = \sigma(a\sigma^{-1}(\cdot) + c) : a \geq 0, c \in \mathbb{R}\}^3$. Apparently, the optimal recalibration function in Equation (41), $h_{f,P}^* \in \mathcal{H}_{\text{logit-normal}}$.

## E.1    Verifying results for UMB

First, we recalibrate $f$ on data distributed as $\mathcal{D}(0.5)$ using UMB.

**Verifying the risk convergence in Theorem 1**    We vary $n \in [10^2, 10^7]$ and $B \in [6, 10^3]$ in the log scale. For each combination of $(n, B)$, we use UMB to recalibrate $\hat{f}$ on data generated from $\mathcal{D}(0.5)$, and compute quadrature estimates of population $R^{\text{cal}}(\hat{h})$, $R^{\text{sha}}(\hat{h})$, and $R(\hat{h})$, as well as their high probability bounds based on Theorem 1. The constant $K$ in Assumption (A3) is selected by numerical maximization as

$$K = \max_{0 \leq z_1 < z_2 \leq 1} \frac{h^*(z_2) - h^*(z_1)}{P[Z \in [z_1, z_2]]}.$$

Figure 1 shows the bounds follow the same trends as their associated population quantities, providing valid upper bounds in all cases.

**Verifying the optimal choice of the number of bins.**    We find empirically optimal $B^{*\text{experiment}}$ that achieves the minimal risk for each choice of $n$. We compute the theoretically optimal choice of the number of bins, $B^{*\text{theory}}$, by minimizing the finite-sample upper bounds. Figure 2 shows $B^{*\text{experiment}}$ follows the same trend with $B^{*\text{theory}}$, both scales in $O(n^{1/3})$.

## E.2    Comparing recalibration methods

To highlight the benefits and drawbacks of UMB's nonparametric nature, we compare UMB with (semi-)parametric recalibration methods in scenarios where the parametric assumption is correct and where it is misspecified. We compare the method under study, uniform-mass binning (UMB), with 3 other recalibration methods: uniform-width binning (UWB) [18], Platt scaling [40][4], and a hybrid parametric-binning method [29]. Note that Platt scaling and the hybrid method adopt the parametric assumption $h^* \in \mathcal{H}_{\text{logit-normal}}$.

For the first setting, we construct optimal recalibration function $h^* \in \mathcal{H}_{\text{logit-normal}}$ so that the parametric assumption of Platt scaling and the hybrid method holds. In particular, we consider the distribution $Z \in \text{Uniform}[0, 1]$ and $Y \mid Z \sim \text{Bernoulli}(h_{\text{logit-normal}}(Z; a, c))$ with $a = 4$ and $c = 0$. For the second setting, we construct $h^* \in \mathcal{H}_{\text{beta}}$ but $h^* \notin \mathcal{H}_{\text{logit-normal}}$ so that the parametric assumption fails. In particular, we consider the distribution $Z \sim \text{Uniform}[0, 1]$ and $Y \mid Z \sim \text{Bernoulli}(h_{\text{beta}}(z; a, b, c))$ with $a = 0.1$, $b = 4$, and $c = 0$. For each setting, we fix calibration sample size to be $n = 5000$.

---

[3]We say $Z \sim \text{Logit-Normal}(\mu, \tau^2)$ if $\sigma^{-1}(Z) \sim N(\mu, \tau^2)$ [1]. Similar to beta calibration [28], we adopt the name "logit-normal calibration" after a simple example: if $Y = \text{Bernoulli}(0.5)$, $Z \mid Y = i \sim \text{Logit-Normal}(\mu_i, \tau_i^2)$ for $i \in \{0, 1\}$, then the optimal recalibration function $\mathbb{E}[Y \mid Z = z] = h_{\text{logit-normal}}(z; a, c)$ for some $a, c$ depending on $\mu_i$'s and $\tau_i$'s.

[4]The original Platt scaling operates on outputs of real-valued SVM outputs [40]. For probabilistic classifiers, we follow [38, 29, 22] and implement Platt scaling by first transforming probabilities onto the real line via the logit transform $\sigma^{-1}$.

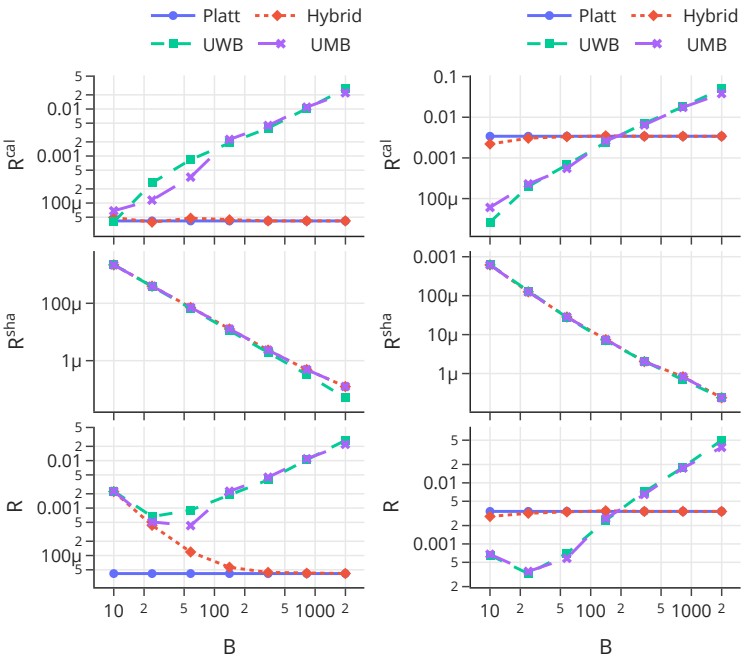

(a) Correct parametric assumption  (b) Misspecified parametric assumption

Figure 5: Risks vs. number of bins $B$.

**Risks as functions of the number of bins** $B$    We traverse the number of bins $B \in [10, 2000]$ in the log scale and compare how each method behaves as $B$ changes. When the parametric assumption is correct, the hybrid method achives significantly lower $R^{\text{cal}}$ and overall $R$ than UMB and UWB for sufficiently large number of bins (Figure 5a), an advantage highlighted in [29]. In contrast, when the parametric assumption fails, the binning methods UMB and UWB has better performance with the optimal number of bins (Figure 5b). This is because Platt scaling and hybrid methods are intrinsicly biased when $h^* \notin \mathcal{H}_{\text{logit-normal}}$, as noted in Section 4.2.

**Quantitative results of risks under optimal** $B$    For each setting, we fix $B$ that achieves low recalibration risk for UWB and UMB in Figure 5. Specifically, we choose $B = 2 \lfloor n^{1/3} \rfloor = 34$ for the correct parametric assumption setting, and $B = \lfloor n^{1/3} \rfloor = 17$ for the misspecified parametric assumption setting. Then, for each setting, we compare the 90% quantiles of risks of each recalibration method fitted on 100 random replicates of calibration datasets of size $n = 5000$.

Table 1 quantititively verifies that Platt and the Hybrid method achieves lower $R^{\text{cal}}$ and overall $R$ if the parametric assumption is correct, and UWB and UMB achieves lower $R^{\text{cal}}$ and overall $R$ when the parametric assumption fails.

**Visualization of calibration curves**    We fix the calibration dataset and visualize the calibration curves for all methods under the two settings. Figure 3 shows that the binning methods (UWB and UMB) closely track the optimal recalibration function $h^*$ in both settings. In contrast, the hybrid approach follows the Platt scaling estimates, leading to an inherent bias from $h^*$ when the parametric assumption is invalid (Figure 3b).

### E.3    Comparing recalibration schemes under label shift

We consider the label shift with source distribution $\mathcal{D}(0.5)$ and target distribution $\mathcal{D}(\pi_Q)$, where $\pi_Q$ varies in $\{0.01, 0.05, 0.1, 0.2, 0.3, 0.4, 0.5\}$. The results where $\pi_Q > 0.5$ can be inferred by symmetry and hence not experimented. We vary $n_P$ in $\{10, 10^3, 10^5, 10^7\}$ and $n^Q$ in $\{10, 10^3, 10^5\}$.

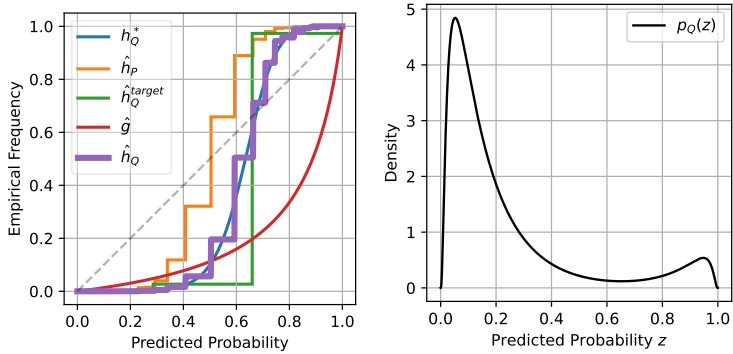

Figure 6: *Left*: calibration curves of for COMPOSITE $\hat{h}_Q$, SOURCE $\hat{h}_P$, TARGET $\hat{h}_Q^{\text{target}}$, and LABEL-SHIFT $\hat{g}$. *Right*: the marginal density of $Z = f(X)$ under $Q$.

Aside from our proposed recalibration function $\hat{h}_Q = \hat{g} \circ \hat{h}_P$ (17), referred to as COMPOSITE, we consider three other calibration approaches as baselines: (1) SOURCE, denoted as $\hat{h}_P$, which is only calibrated on the source data, (2) LABEL-SHIFT, denoted as $\hat{g}$, which performs label shift correction without calibration, and (3) TARGET, denoted as $\hat{g}_Q^{\text{target}}$, which is only calibrated on the target data. The number of bins $B$ are chosen to be $n_P^{1/3}$ for COMPOSITE and SOURCE, and $n_Q^{1/3}$ for TARGET.

Table 2 shows the risks for different approaches with $\pi_Q = 0.1$, $n_P = 10^3$, and $n_Q = 10^2$. In terms of $R^{\text{cal}}$, COMPOSITE performs the best, as it is calibrated to the target distribution by taking advantage of the abundant source data. In terms of $R^{\text{sha}}$, LABEL-SHIFT achieves $R^{\text{sha}} = 0$ due to the strictly increasing $\hat{g}$, but it suffers from high $R^{\text{cal}}$. COMPOSITE and SOURCE achieve smaller $R^{\text{sha}}$ than TARGET, as a result of using more bins on a larger sample. Considering the combined impact of calibration and sharpness, our approach COMPOSITE attains the lowest overall recalibration risk $R$ as well as MSE.

Figure 6 shows the optimal recalibration function $h^*$ and the recalibration functions for the four approaches. It can be seen that COMPOSITE best estimates $h^*$ with the highest resolution.

