# OpenReview forum: "Minimum-Risk Recalibration of Classifiers"
_NeurIPS.cc/2023/Conference — NeurIPS 2023 spotlight_

### Official Review · Reviewer_GPZN · 2023-07-06

**Soundness:** 4 excellent
**Presentation:** 3 good
**Contribution:** 2 fair
**Rating:** 6
**Confidence:** 4

**Summary:**

Background:
Generating reliable probability estimates alongside accurate class labels is crucial in classification tasks. Calibration, which refers to the alignment between predicted probabilities and empirical frequencies of labels, is highly desirable in various applications. However, many machine learning algorithms lack inherent calibration.

This paper aims to address these issues in two ways:
1. Development of a unified framework for recalibration that incorporates both calibration and sharpness in a principled manner.
2. Proposal of a composite estimator for recalibration under label shift, which converges to the optimal recalibration and enables sample-efficient adaptation of classifiers to label-shifted domains.

**Strengths:**

1. This paper is exceptionally well-written, with a clear and logical structure, making it easily understandable to a wide audience.
2. The theoretical foundation of this paper is solid, the proof process is very detailed, making it easy for people to follow.
3. The algorithm proposed by the author is simple and efficient, making it easy to implement.

**Weaknesses:**

1. Although the author provides solid theoretical analysis, this paper lacks enough experiments to demonstrate the effectiveness of their method.
2. The author employs three assumptions, and in my view, assumption 2 is a rather strong one, which is difficult to guarantee in practical applications.
3. The experiments in this paper were conducted primarily on toy datasets, and the author do not provide any experiments on how to apply Recalibration in real-world applications.

**Questions:**

1. In Eq. (9), I understand that the function $\hat{h}(z)$ calculates the expected label within the bin that $z$ belongs to. So, why is $\hat{h}$ a monotonically increasing function of $z$ and its growth pattern similar to the cumulative distribution function (CDF) of $z$? Although you have made Assumption 2, can I understand that this assumption implies that $y$ must follow a specific distribution form in order to satisfy the monotonicity of $\hat{h}$?
2. In the experimental section, the author only presents a very simple toy scenario, and I am quite curious about the performance of the author's algorithm in more complex scenarios. For instance, in real-world classification datasets like cifar-10-long-tail or cifar-100-long-tail, how does the recalibration effect of more complex models, such as neural networks or decision trees?
3. In Fig. 1 (a), as the value of $n$ increases, regardless of the magnitude of $B$, $R^{cal}(\hat{h})$ exhibits a noticeable decrease. Does this imply that when the dataset size $n$ becomes sufficiently large, $R^{cal}(\hat{h})$ has already reached a level where recalibration is unnecessary?

If the author can answer my question, I will change my score.

**Limitations:**

yes

---

> ### Author Rebuttal · Authors · 2023-08-09
>
> Thank you for the valuable feedback on our work. With gratitude for the positive evaluation, we are committed to further clarifying our contributions by addressing the concerns and questions raised. To streamline this endeavor, our response is organized to initially address the highlighted weaknesses, followed by detailed point-by-point responses to the specific questions.
>
> ### **Weaknesses**
> #### 1\. Insufficient experiments:
> We believe that the numerical evidences presented in our original submission effectively validate both the theoretical claims and the efficacy of our method. Specifically, we validate the main results outlined in Section 4, including the risk upper bounds in Theorem 1 (Figure 1) and the optimal choice of the number of bins $B$ (Figure 2). Additionally, the significant contributions of Section 5, such as the explicit form of optimal recalibration under label shift and a two-step recalibration estimator with theoretical guarantees, are substantiated by Table 1.
>
> While we maintain confidence in the sufficiency of our original numerical experiments to validate our theoretical findings, we also acknowledge the value of extending numerical studies, e.g., by including comparisons to other methods. To address this, we have conducted additional numerical experiments that compare the performance of our proposed UMB method against three other approaches: UWB [16], Platt scaling [34], and the hybrid method [26]. The results of these supplementary comparisons are included in the Author Rebuttal report.
>
> #### 2\. Validity of Assumption 2:
> We appreciate the reviewer's recognition of the potential stringency and practical challenges related to ensuring Assumption 2. However, it's important to highlight that while this assumption might appear stringent, it remains sensible for a reasonably well-performing classifier $f$, for which the underlying trend is expected to persist even if predicted probabilities $f(x)$ are imprecise. Specifically, we should expect $f(x_1)\leq f(x_2)$ when $P[Y=1|x_1]\leq P[Y=1|x_2]$ for a reasonable classifier.
>
> Additionally, it's worth noting that similar monotonicity assumptions are prevalent in related literature. For instance, such assumptions are utilized in recalibration through isotonic regression [44], maintaining accuracy via order-preserving maps [45], and selecting bin numbers for monotonicity preservation [37].
>
> #### 3\. Lack of experiments with real-world datasets:
> Our main emphasis in this paper lies in presenting a fresh recalibration perspective alongside a thorough analysis of a working method, making it primarily theoretical in nature. Although the application of our framework to real-world contexts holds potential interest, we believe its incorporation is not essential within the scope of the present paper. As such, we regard this as a captivating and promising avenue for future endeavors—to extend our comprehensive framework, encompassing both calibration and sharpness risks, to real-world applications.
>
> ### **Questions**
> #### 1\. Monotonicity of $\hat{h}$?
> First of all, we want to clarify that Assumption (A2) enforces the monotonicity of $h^*$, NOT imposing the monotonicity of $\hat{h}$. As the reviewer observed, $\hat{h}$ is an estimate of $h^*$ computed as the expected label within the bin that $z$ belongs to, and may NOT be monotone due to the finite sampling effect. However, we anticipate concentration of $\hat{h}$ to $h^*$, especially with narrow bins and ample per-bin data, leading to monotone increasing $\hat{h}$ with high probability.
>
> Furthermore, the reviewer accurately notes that Assumption (A2) concerns the distribution of $Y$. Specifically, the monotonicity of $h^*$ imposes a monotonicity requirement on the conditional distribution of $Y$ given $Z=f(X)$; recall $h^*(z)=E[Y|Z=z]$ from Eq. (7).
>
> #### 2\. Applications to more complex scenarios:
> 1. **Real-world data:**
> We agree with the reviewer that applications of our framework to real-world classification tasks would be interesting. Nevertheless, it's important to highlight that many real-world tasks involve **multi-class** classification. While it is conceivable to address this, e.g., by transforming it into multiple pairwise binary classification tasks, multi-class classification scenarios lie beyond the scope of the current paper. Thus, we regard it as an intriguing direction for future exploration.
>
> 2. **Complex models:**
> Post-hoc recalibration treats probabilistic classifiers as black-box entities, only utilizing their predicted probabilities as inputs. Therefore, our recalibration framework applies to **any** probabilistic classifiers, including neural networks and decision trees. As far as we understand, the performance of the model and recalibration are not directly linked. Although different models might yield distinct joint distributions of probabilistic predictions $f(X)$ and labels $Y$, indirectly influencing the fitted recalibration function $\hat{h}$, its performance will remain consistent if it holds distribution-free calibration guarantees [18].
>
> #### 3\. (Un-) necessity of recalibration for large $n$?
> We appreciate the reviewer's keen observation regarding the decreasing trend of $R^{cal}(\hat{h})$ as $n$ increases in Fig. 1(a). This behavior signifies that $\hat{h}$ effectively calibrates $f$, resulting in a well-calibrated composition $\hat{h}\circ f$ for large $n$; it's important to clarify that this doesn't necessarily imply $f$ is inherently well-calibrated. Hence, if the initial classifier $f$ lacks calibration, it remains relevant to estimate a post-hoc recalibration function $\hat{h}$. Additionally, we'd like to emphasize that the aspect of sharpness is entirely overlooked in this context; when $B=1$, the composition $\hat{h}\circ f(x)\to EY$ (for all $x$) as $n\to\infty$, leading to a constant classifier that's calibrated but devoid of information.

---

### Official Review · Reviewer_yKq4 · 2023-07-07

**Soundness:** 4 excellent
**Presentation:** 4 excellent
**Contribution:** 3 good
**Rating:** 8
**Confidence:** 4

**Summary:**

The paper studies a very relevant problem of post-hoc calibration in probabilistic classifiers. There is a great body of work on calibrating probabilistic classifiers so that the predicted probabilities match the empirical label frequencies in the popular machine learning literature. However, calibrating probabilistic classifiers so that they retain their sharpness / refinement is not studied much. The paper focuses on the post-hoc calibration with retained sharpness properties from first principles. Its entry point is the classical decomposition of mean squared error of the classifier into sharpness and (mis)calibration. Traditionally, both the measures were important, but the works in machine learning literature has mostly focussed on (mis)calibration only. There are important contributions in this paper: a) proposing recalibration risk measure which attains its minimum value only when both sharpness and the calibration requirements are met. b) uniform mass binning algorithm to approximate the optimal recalibration map. c) Theoretical results on the binning scheme and binning scheme to bound both the calibration risk and the sharpness risk. d) Formal result on the trade-off between calibration and sharpness. e) recalibration studies in a simple (but important) distribution shift with label shift. All of these results are for binary classification setting.

**Strengths:**

1. It is clear that the paper makes theoretical contributions to an important problem. It has been observed that major recalibration algorithms in machine learning literature not just ignore the sharpness measure, but they degrade it [1]. There is a trade-off between these quantities, and this paper formally state this with some actionable insights to balance this trade-off.
2.  Theorem 1 generalises a previous result in the literature with an additional insight on the sharpness risk. Generalisations and connections to previous results is a sign of great work.
3. While the results on label shift are easy consequences of the general recalibration results presented earlier in the paper, it is still great to see this extension.
4. In my opinion, the paper is significant and relevant to the machine learning community. I believe the paper will spur more work on post-hoc calibration methods.
5. The paper is excellently written, easy to read and understand.





[1] Aditya Singh et al. On The Dark Side Of Calibration For Modern Neural Networks. (2021) (http://www.gatsby.ucl.ac.uk/~balaji/udl2021/accepted-papers/UDL2021-paper-074.pdf)

**Weaknesses:**

I do not have major concerns with this paper. I have some questions though (please see below):

**Questions:**

1. The optimal recalibration function is same as the canonical calibration map defined in [2] (Section 3.1). While it is clearly interesting to see that a canonical calibration map is the optimal recalibration map, could authors comment on this connection? The optimal recalibration map is certainly a calibrated function (as the Proposition 1 in [2] states), I am curious about some insights on why it would also be a map to minimise the sharpness risk.

2. Additionally, [2] also propose a sort of general methodology to estimate this canonical calibration map (Section 4.1). While they do not comment on the partitioning scheme, it would be interesting to draw insights on general partitioning schemes and the bound on the recalibration risk.

3. While the authors mention that extending the results presented in the paper to multi-class setting is a future research direction, could authors see that the estimator provided in [2] could be useful for multi class case (as [2] do not restrict itself to binary classification)?


[2]  Juozas Vaicenavicius et al. Evaluating model calibration in classification. (2019)

**Limitations:**

Obviously, there are limitations to this work, as there are with any works. However, the authors have been very clear in stating them.

---

> ### Author Rebuttal · Authors · 2023-08-09
>
> We would like to express our sincere gratitude for the comprehensive understanding and the positive evaluation from the reviewer, especially in acknowledging our theoretical contribution to an important field while providing actionable insights, and recognizing the impact of this work on the community.
> We also appreciate the invaluable questions and suggestions from the reviewer, which has led us to think deeper in terms of extending our theory to the multi-class scenario while drawing connections from a previous paper [41] (we use the reference number in the paper to avoid confusion).
> We present our perspectives as follows and hope these will address the reviewer's questions and lead to insightful discussions.
>
> ### **Questions**
> #### 1\. Connection with the canonical calibration map in [41]:
> The reviewer's observation is accurate; the optimal recalibration function in our work is the same as the canonical calibration function defined in Eq. (4) of [41] in the binary classification setting. Below, we elaborate on this connection in three aspects.
>
> 1. **Equivalence:**
> In the framework of [41], a classifier $f: \cal{X} \to \cal{Y}$ is considered *reliable* if and only if $P[Y\in\cdot|f(X)]=f(X)$, that is, the conditional distribution of the target class, given any prediction made by $f$, precisely matches that prediction. This aligns directly with the notion of a *(perfectly) calibrated* classifier, as defined in our paper through Definition 1.
>
> 2. **Calibration:**
> As pointed out by the reviewer, the optimal recalibration function is calibrated as it follows the form in Proposition 1 in [41].
>
> 3. **Sharpness:**
> In our work, we define the optimal recalibration function through the minimum risk criterion, requiring the achievement of zero recalibration risk. Remarkably, this implies the optimal recalibration function has 0 calibration risk and 0 sharpness risk according to Proposition 1. Alternatively, we can also examine the definition of the sharpness risk (Definition 4) for the optimal recalibration function $h^*(z)=\mathbb{E}\left[ Y \mid f(X) = z\right]$. Observing $h^*(f(X))$ as a function of $f(X)$ and applying towering property, we have
> \begin{align}
>     \newcommand{\EP}{\mathbb{E}}
>     \EP \left[ Y \mid h^*(f(X)) \right]
>     = \EP \left[ Y \mid \EP \left[Y \mid f(X) \right] \right]
>     = \EP \left[ \EP \left[Y \mid f(X) \right] \mid \EP \left[Y \mid f(X) \right] \right]
>     = \EP \left[Y \mid f(X) \right],
> \end{align}
> which establishes that $R^{sha}(h^*) = 0$.
>
>     Intuitively, $h^*$ takes the form of expectation of $Y$ conditioned on the full prediction $f(X)$, in a sense carrying all the "information" within $f(X)$ about $Y$. Therefore, using $h^* \circ f$ will not reduce the explained variance by $f$, thus achieving 0 sharpness risk.
>
>
>
> #### 2\. Exploring insights from partitioning schemes in [41]:
>
> The reviewer has identified a major challenge for recalibration using a general partitioning scheme.
> In particular, the consistency results presented in Theorem 1 of [41] lay a foundation for estimating calibration error, prompting the question of deriving convergence rates for calibration risks. However, effectively controlling the sharpness risk can be intricate when an explicit description of a partitioning scheme is absent.
> One insight we draw from [41] is that the maximum diameter of sets in the partition can be used to effectively measure the granularity of the partition, which may be useful in developing sharpness risk bounds.
> Considering these aspects, we recognize that bounding the recalibration risk for overall recalibration risk for general partitioning schemes is a promising, albeit challenging, trajectory for future research endeavors.
> We appreciate the reviewer's thought-provoking comment, as it has motivated us to delve into the intricacies of general partitioning schemes and derive these valuable perspectives.
>
>
> #### 3\. Potential usefulness of partitioning method for multi-class settings:
>
> We perceive the partition-based estimator in [41] as a promising candidate for extending our framework to multi-class scenarios, representing a natural multi-dimensional progression from the histogram-binning estimator. Beyond the specific estimator outlined in that work, we also recognize an additional potential application of [41] stemming from its introduction of a calibration lens. This lens accommodates diverse facets of partial calibration, a concept of particular interest within the realm of multi-class classification [16, 19].
>
> Furthermore, the work by [41] establishes the almost sure consistency of a binned (=partitioned) estimator for expected miscalibration through Theorem 1. This achievement could potentially serve as a foundational step in establishing the convergence of the binned estimator's calibration risk towards zero, offering a promising initial stride in this direction.

---

> > ### Comment · Reviewer_yKq4 · 2023-08-17
> > **Post rebuttal comment**
> >
> > Thanks to authors for the detailed response. I'm certainly glad that authors found my comments useful, and have been able to draw further connections. Unfortunately, I haven't been able to engage in the discussion to the capacity that I'd have hoped due to some personal issues, but I find the response very insightful. I remain confident that the current paper should appear at NeurIPS, and hence I'm also increasing my score.

---

### Official Review · Reviewer_USpM · 2023-07-09

**Soundness:** 4 excellent
**Presentation:** 3 good
**Contribution:** 4 excellent
**Rating:** 7
**Confidence:** 3

**Summary:**

This paper introduces the concept of minimum risk recalibration, utilizing Mean Squared Error (MSE) decomposition. The authors provide justification for their approach by demonstrating that minimizing the proposed risk yields simultaneous minimization of the calibration risk while preserving the sharpness of the probability forecaster. Furthermore, the authors employ the MSE decomposition to analyze the recalibration method known as UMB. Theoretical analysis reveals that selecting an appropriate number of bins allows for achieving a balance between the calibration risk and sharpness. Expanding on their findings, the authors apply their methodology to address the problem of recalibration in label shift. They provide theoretical and experimental validation for their approach in the context of label shift, further supporting the effectiveness of their proposed method.

**Strengths:**

1. The authors provide a rigorous introduction to key statistical measures essential for the recalibration task, such as recalibration/calibration risk and sharpness risk. They demonstrate that minimizing these measures can result in a well-calibrated forecaster.

2. Building upon the introduced statistics and their properties, the authors enhance the analysis of UMB. They unveil a tradeoff between calibration risk and the preservation of sharpness by establishing a high-probability error bound. Furthermore, they leverage this upper bound to guide the selection of the number of bins in the UMB method, highlighting the practical significance of the proposed bound and decomposition.

3. The authors extend the application of their proposed method to address the challenge of label shift, showcasing the validity of their work in practical downstream tasks.

4. The paper thoroughly discusses the mildness of the assumptions made and experimentally validates the non-trivial nature of the proposed bounds and schemes.

**Weaknesses:**

While the outcome of this study is fruitful, it is encouraged that the authors delve into a further discussion regarding the potential extension of their framework to the multi-class scenario. This extension would provide additional intuitive information and enhance the applicability of their proposed methodology.

**Questions:**

Please refer to the Weakness part.

**Limitations:**

The authors have analyzed the limitation of the used method in the appendix.

---

> ### Author Rebuttal · Authors · 2023-08-09
>
> We are overwhelmingly grateful for the reviewer's in-depth understanding and unreserved recognition of our work, encompassing nontrivial concepts, rigorous analysis, practical significance of the tradeoff, applications to downstream tasks, assessment of assumptions, and experiment design for theory verification.
> We indeed agree with the reviewer that one of the most promising directions forward is to extend our framework to the multiclass scenario.
> We are delighted to provide further discussion in this paper to enhance intuitive understanding and practical applicability of our methodology.
>
> In the context of multi-class classification, the concept of calibration takes on various forms [19].
> An intriguing avenue for extension involves canonical calibration [41], which directly extends our methodology.
> A major challenge of the extension lies in designing the binning scheme in a multidimensional space, a complexity that has been explored in [19].
> While calibration guarantees have been explored in [35], establishing sharpness risk bounds within this multi-class context remains an engaging pursuit.
> The interplay between calibration and sharpness could potentially guide the development of a binning strategy that balances these aspects in multi-class classification.
>
> We extend our appreciation to the reviewer once more for this invaluable suggestion of including additional discussion on the multi-class scenario.
> Furthermore, we would appreciate the reviewer's further input that could enrich this discussion and enhance the value of our work.

---

> > ### Comment · Reviewer_USpM · 2023-08-20
> >
> > Thanks to the authors for providing related works in the multi-class scenario. I will keep my score since the difficulty of the extension to the multi-class scenario is inherent in the task of probability calibration and does not weaken the contribution of this work.

---

### Official Review · Reviewer_kAN5 · 2023-07-25

**Soundness:** 3 good
**Presentation:** 4 excellent
**Contribution:** 3 good
**Rating:** 6
**Confidence:** 1

**Summary:**

This work proposes a calibration method for probabilistic classifiers. It is known that most machine learning models produce predictions with high confidence that yield a distribution different than the underlying true label distribution. The proposed method focuses on calibration without a loss on the prediction performance and is claimed to adjust for label shift.

**Strengths:**

- This paper is written clearly with good and easy-to-follow notations. The different definitions are well-placed and help the reader to follow the flow of the paper. Overall good structure of the paper.
- The equal focus during the calibration on both calibration and sharpness goals is well-motivated and importantly this setting. The theory is developed rigorously and the different choices and assumptions are justified.
- This paper extends the proposed work to label-shfit setting and provides a good discussion on "recalibration under label shift" .
- This paper provides experimental results that show the validity of the theoretical work proposed.

**Weaknesses:**

I think this paper lacks the following two points:

- A discussion in section 1.1 highlighting to the reader how this paper approaches the recalibration problem differently than existing work. Only the work in [26] has been addressed.
- Section 6 does not include a discussion about the empirical results of previous works.

**Questions:**

N/A

---

> ### Author Rebuttal · Authors · 2023-08-09
>
> Thank you for the valuable feedback on our work. With gratitude for the positive evaluation, we are dedicated to clarifying and enhancing our contributions by addressing the raised concerns.
>
> ### **Weaknesses**
> #### 1\. Insufficient discussion on distinction from existing work:
> The reviewer's observation is valid, and we appreciate the opportunity to address this concern. While we have selected [26] for the main comparison, it's important to note that our engagement with existing work extends beyond this comparison. In Section 4, alongside the comparison with [26], we underscore in Remark 1 that our calibration risk bound aligns with [18] up to a constant factor in the failure probability. In Section 5, we compared target sample complexity with [28,2,12] in Remark 5, pointing out that our method, which only uses target labels, achieves the same order of risk bounds as the methods using only target features.
>
> We would like to provide further insight into our rationale for selecting [26,18] as the basis of comparison in Section 4. Our objective is to holistically and quantitatively address both calibration and sharpness preservation. A main contribution of our work is to develop risk bounds for both calibration and sharpness risks, which subsequently inform the optimal choice of bin number in uniform mass binning -- a recalibration method extensively studied theoretically and applied in practice. To the best of our knowledge, [26,18] represent the most competitive theoretical works that are relevant to our goal, as they provide state-of-the-art calibration error bounds under their specified assumptions. In contrast, [32,37] lack theoretical underpinning and primarily focus on empirical assessment.
>
> We value the reviewer's perspective feedback and appreciate the opportunity to clarify our choices of baseline works. Should there be additional works that are better suited for comparison with our theoretical framework, we would appreciate the reviewer's further input, which would improve the robustness and comprehensiveness of our work.
>
> #### 2\. Lack of empirical results comparison and discussion with prior works:
> We greatly appreciate the reviewer's observation, which makes us realize the importance of including empirical comparisons between the proposed methods and previous works.
> In addition to our original experiments which validated our theoretical claims, we have conducted additional numerical experiments to compare the performance of the UMB method with three other approaches—UWB [16], Platt scaling [34], and hybrid method [26]. The results and discussion of these comparisons have been incorporated into the Author Rebuttal report. This inclusion enhances the comprehensive evaluation of our work and strengthens its practical implications.

---

> > ### Comment · Reviewer_kAN5 · 2023-08-18
> >
> > Thanks to the authors for addressing my concerns. I will keep my score as I am unfamiliar with the overall related work as shown in my confidence score.

---

### Official Review · Reviewer_a48G · 2023-07-27

**Soundness:** 3 good
**Presentation:** 3 good
**Contribution:** 2 fair
**Rating:** 6
**Confidence:** 3

**Summary:**

This work looks at methods for calibrating probabilistic classifiers for Mean Squared Error by decomposing it into calibration error and sharpness errors. It gives finite sample error guarantees on both of these for the Uniform Mass Binning method. By balancing sharpness and calibration error, they also propose the optimal number of bins to use. Additionally, they look at the problem of calibrating classifiers in the case of label distribution shift between train and test. Their results show that transferring a calibrated classifier requires significantly fewer target samples compared to recalibrating from scratch. They validate our theoretical findings through numerical simulations.

**Strengths:**

- Existing works have looked at finite sample bounds for the calibration error but not the sharpness (as mentioned in this paper). This is the first method to look at both sharpness and calibration together which is an important criterion.
- The paper is generally well written.

**Weaknesses:**

- The main contribution is to give risk bounds for sharpness along with the calibration error. In my opinion, this is not a significant contribution. There is not any particular strategy proposed using this analysis but only choosing the number of bins.
- The second problem of using this method for label distribution shift also seems straightforward and I have also asked a related question below.

**Questions:**

- Why did the authors choose to analyze only Uniform Mass Binning and not other methods like Uniform width binning?
- For the label distribution shift problem, is the proposed method equivalent to just scaling all the predicted probablities according to the target distribution and then using the uniform binning technique?
- In definition 4, the sharpness risk of h over f, is this the sharpness risk of h.f - sharpness risk of f?
- Why is Assumption 3 needed for the label distribution case?
- Just to confirm, is this the first sharpness risk bound?
- In figure 2a, why is the gap increasing between the theoretical and empirical bound as n increases?
- In table 1, the different methods compared are not clear to me. Can the authors please explain?

**Limitations:**

Yes

---

> ### Author Rebuttal · Authors · 2023-08-09
>
>
> We appreciate the valuable comments and feedback. Our response is structured to address the highlighted weaknesses first, followed by detailed point-by-point responses to specific questions.
>
> ### **Weaknesses**
>
> 1\. Not a significant contribution:
>
> We appreciate the reviewer's recognition of our paper's unique contribution in jointly addressing sharpness and calibration, introducing a new recalibration perspective. Yet, we respectfully disagree that our contributions lack significance. We elaborate on our work's importance as follows:
> 1. **Holistic approach:**
> Post hoc recalibration is a crucial issue with a long history. Our paper advances the recalibration field by addressing sharpness and calibration together, bridging a gap in prior research. Sharpness, though often overlooked, plays a crucial role in recalibration. Relying solely on the calibration criterion may lead to suboptimal recalibration. By considering both aspects, we provide a comprehensive and principled framework to effectively tackle the problem, a notable advancement.
> 2. **Foundational understanding:**
> Although not novel, histogram binning presents a functional approach that lays a foundational step in illuminating the recalibration perspective. Our aim is not to propose a complex or competitive strategy, but to establish a solid theoretical basis for recalibration. This foundational understanding holds intrinsic value, complementing practical strategies. Moreover, the simplicity of our method is not a drawback, but rather a blessing, as it extends practical applicability.
> 3. **Bin selection significance:**
> We acknowledge the importance of bin selection in histogram binning literature, a well-recognized topic in calibration research. Our approach aligns with recent efforts, such as [26] and [37], which emphasize the need for improved binning strategies. Our paper contributes by proposing a theoretically-backed approach for selecting the optimal number of bins, bolstering recalibration reliability.
>
> We believe these contributions significantly enhance recalibration and pave the way for future advancements.
>
> 2\. Perceived simplicity of the method:
>
> The reviewer described scaling predicted probabilities via the label shift formula followed by recalibration through uniform mass binning (UMB) in Question-2. We stress a key distinction: the operation order is reversed compared to Eq. (15) or (17) in the original submission. Thus, the reviewer's method may not necessarily estimate the optimal recalibration function under label shift. Recall that our recalibration function's optimality is substantiated by Theorem 2's risk bounds.
>
> We appreciate the reviewer's engagement and hope this response clarifies our method's nuances and strengths. Also, we reiterate that the method's simplicity underscores its broad applicability and foundational significance, rather than being a drawback.
>
> ### **Questions**
> 1\. Rationale for UMB:
>
> Our choice of UMB over UWB was driven by analytical considerations. UMB's well-balanced binning property (Lemma 3 in Appendix B.1), ensuring nearly equal sample sizes per bin, aids analysis. In contrast, UWB can yield varied per-bin sample sizes depending on distributions. Consequently, UWB's calibration risk bounds conservatively rely on the smallest sample sizes among bins [17, Corollary 3], whereas UMB provides distribution-free calibration risk bounds. Moreover, UMB's well-balanced binning ($\Phi_{balance}(B,\alpha)=1$ in Lemmas 6 and 7) ensures robust sharpness risk bounds across diverse distributions.
>
> 2\. Clarification of the method:
>
> Please see the response to Weakness-2.
>
> 3\. Definition 4:
>
> We presume that the reviewer inquires if the sharpness risk of $h$ equals the reduction in sharpness of $f$ resulting from the application of $h$, i.e., sharpness of $f$ minus sharpness of $h\circ f$, which an accurate interpretation (see Line 146-147).
>
> 4\. Assumption 3:
>
> Assumption (A3) is optional, and its inclusion enhances the sharpness risk bound from $O(1/B)$ to $O(1/B^2)$ (Theorem 1). For the label shift case (Theorem 2), omitting Assumption (A3) relaxes the sharpness risk bound term from $\frac{8K^2}{B^2}$ to $O(1/B)$. We opt for (A3) whenever feasible to showcase the tightest upper bound within a context reasonably common in practical scenarios.
>
> 5\. First sharpness risk bound:
>
> To our knowledge, this indeed represents the first formulation of a sharpness risk bound. It's worth mentioning that a related study by [26] found that MSE only increases by O(1/B) by discretization (Proposition D.4, [26]), which bounds sharpness risk by O(1/B) for recalibration functions with 0 calibration risk. In our work, we integrate this insight into Theorem 1, asserting $GRP(\hat{h})\leq\frac{2}{B}$ in the absence of Assumption (A3).
>
> It's important to underscore that [26] didn't explicitly label their excess MSE bound as a sharpness risk bound, nor did they emphasize the independent existence of such a bound beyond discretization. In fact, the bound of sharpness risk strictly implies their results as a special case. Furthermore, we enhance the sharpness risk bound to $O(1/B^2)$ by introducing Assumption (A3).
>
> 6\. Figure 2a:
>
> Our focus is mainly on the order of the risk bound, which holds significance in asymptotic contexts. As such, the expanding gap in Figure 2-a and the uniform gap in Figure 2-b could potentially be attributed to a constant multiplicative factor. Remarkably, the theoretical and empirical optimal values for $B$ closely align in order, evident from their near-parallel trends in Figure 2(b).
>
> 7\. Table 1:
>
> Due to page limit, we moved the detailed method explanation to Appendix E.2. We're aware of possible clarity concerns in Table 1's caption and understand the value of summarizing these details in the Experiments section. In the forthcoming camera-ready version upon acceptance, we aim to improve clarity by incorporating these explanations into Table 1's caption or integrating them into the main text.

---

> > ### Comment · Reviewer_a48G · 2023-08-19
> > **Response**
> >
> > I would like to thank the authors for providing the detailed response. After reading the rebuttal and other reviewers' comments, it seems like providing guarantees for sharpness and a strategy to choose optimal bins is an important contribution to a fundamental problem. I have  increased my score.

---

### Author Rebuttal · Authors · 2023-08-10

We appreciate the reviewers' valuable feedback. Our Author Rebuttal addresses recurring themes, offering clarification on methodological contributions and presenting supplementary numerical evidence. Comprehensive point-by-point responses are available in individual rebuttals.

### **Summary of contributions**
**Key highlights.**
Allow us to succinctly underscore the key contributions in this paper.
1. We introduce a novel quantitative framework that interlaces both sharpness and calibration, thereby yielding further insights for the recalibration problem.
2. Applying this framework to histogram binning method, we obtain finite-sample error bounds for the calibration risk and the sharpness risk.
3. Our analysis illuminates a principle guiding the selection of an optimal number of bins, notably identifying $B = O(n^{1/3})$ where $n$ is the calibration dataset size.
4. Considering the label shift, an exemplar of distributional shift, we identify the optimal recalibration function while proposing a pragmatic two-step estimator, fortified with a convergence guarantee.

We appreciate the recognition of our work's theoretical significance by Reviewers 2, 3, 4, and 5. Reviewer 1 also partially acknowledged this contribution, with specific inquiries that we addressed comprehensively in our individual response.

**Significance of contributions.**
We further elaborate on the importance of our work as follows.
1. **Holistic approach:**
Our paper makes a distinct contribution by addressing both sharpness and calibration concurrently, bridging a gap that has persisted within recalibration research. The interplay between these elements, often overlooked, profoundly influences recalibration outcomes. By holistically integrating these aspects, we introduce a comprehensive and principled solution, marking a substantial advancement.
2. **Foundational insight:**
Our proposed method based on histogram binning -- a pragmatic solution among potential alternatives -- and its analysis illuminate a novel recalibration perspective. We underscore that this paper is primarily aimed at establishing a conceptual framework underpinned by strong theoretical foundations. This foundational comprehension carries intrinsic value, effectively complementing practical strategies. Additionally, the simplicity of our approach extends its applicability rather than serving as a limitation.
3. **Significance of optimal bin selection:**
We acknowledge the significance of bin selection, in line with recent work in histogram binning literature such as [26] and [37], emphasizing the need for improved strategies and metrics. Our contribution offers a theoretically-supported approach for optimal bin number selection, bolstering the reliability of recalibration outcomes.

### **Additional experiments**
In response to the reviewers' requests, we conducted additional experiments to compare our proposed method with prominent approaches from the literature. The results of these experiments are detailed in the attached report. Specifically, we assess the performance of four methods: our method (UMB), uniform width binning [16], Platt scaling [34], and Platt-binning [26]. These methods represent two alternative binning approaches, a parametric method, and a hybrid parametric-binning approach, respectively. Additionally, we examine two distinct data distributions that were studied in [25]: Logistic calibration and Beta calibration. The outcomes of these supplementary experiments provide further support for the assertions we made in Section 4.2 of our original submission, where we theoretically compared risk bounds.

Table R.1 presents quadrature estimates of population risks for the four methods under (a) Logistic calibration and (b) Beta calibration. Notably, Table R.1a demonstrates that Platt-binning outperforms scaling and binning methods in terms of $R^{cal}$ when the underlying model assumption is correct (Logistic model). This performance superiority can be attributed to the accurate parametric model assumption, resulting in lower sample complexity compared to nonparametric methods. This finding validates a key claim made in [26]. However, as depicted in Table R.1b, this advantage of Platt-binning diminishes when the true recalibration function deviates from the parametric family. In such cases, the nonparametric binning methods (UMB \& UWB) emerge as the top performers among the four methods.

Figure R.1 provides a visual representation of the recalibration curves for the optimal recalibration function $h^*$ and its estimates under the two distributions. Remarkably, in both Logistic calibration and Beta calibration, histogram binning methods (UWB and UMB) closely track the $h^*$ curve. Conversely, the scaling-binning approach follows the Platt scaling estimates, leading to an inherent bias from $h^*$ in Beta calibration. This visualization provides insightful evidence of the efficacy of our proposed methods.

### **Promising future research: extension to multi-class classification**
As mentioned in the original Discussion, extending our framework to multiclass classification will be an exciting and logical progression. We notice that many reviewers share this view and express interest in its application to multiclass scenarios.

However, a major challenge of the extension lies in designing the binning scheme in a multidimensional space, a complexity that has been explored in [19]. Reviewer 4 (yKq4) highlighted potential insights from partition-based results in [41] for this extension. We acknowledge that while some calibration guarantees have been explored in [35], establishing sharpness risk bounds within this multi-class context remains an engaging pursuit. We believe the interplay between calibration and sharpness could potentially guide the development of a binning strategy that balances these aspects in multi-class classification.

In conclusion, we recognize the multiclass extension as a promising direction for future research.

---

### Decision · Program_Chairs · 2023-09-21

**Decision:**

Accept (spotlight)

**Comment:**

This paper presents a principled approach for recalibrating probabilistic classifiers, with guarantees on both calibration and sharpness. This is in contrast to the existing literature, which mainly focuses on calibration, often at the expense of degradation in sharpness. The authors apply their results the classical histogram binning based calibration, and provide a theoretically sound approach for bin selection, and also discuss an application to recalibration under label shift. Supporting experiments on synthetic data are provided.

The reviewers agree that the paper very well-written and makes solid theoretical contributions. Despite concerns about the paper not providing experiments on real-world data, the general consensus is to accept the paper.

The authors are strongly encouraged to include the following in the camera-ready version:
- detailed discussion of how the paper differs from prior work (Reviewer kAN5)
- a discussion on extension to multi-class problems (Reviewer USpM)
- additional experiments presented as a part of the rebuttal (Reviewer GPZN)